

# A meteorological and blowing snow dataset (2000-2016) from a high-altitude alpine site (Col du Lac Blanc, France, 2 720 m a.s.l.)

Gilbert Guyomarc'h[1] [*], Hervé Bellot[2], Vincent Vionnet[1,3], Florence Naaim Bouvet[2], Yannick Déliot[1], Firmin Fontaine[2], Philippe Puglièse[1], Mohamed Naaim[2], Kouichi Nishimura[4]

[1] Météo-France - CNRS, CNRM – UMR 3589, CEN, St-Martin d'Hères, France

[2] Univ. Grenoble Alpes, IRSTEA, UR ETNA, F-38042 St-Martin-d'Hères, France

[3] Centre for Hydrology, University of Saskatchewan, Saskatoon, SK, Canada

[4] Graduate School of Environmental Studies, Nagoya University, Nagoya, Japan

*Now at: Météo France, DIRAG, Point à Pitre, Guadeloupe, France

Correspondence to: Florence Naaim-Bouvet (florence.naaim@univ-grenoble-alpes.fr)

**Abstract.**

A meteorological and blowing snow dataset issued from the high-altitude experimental site of Col du Lac Blanc (2720 m
altitude, Grandes Rousses mountain range, France) is presented and detailed in this paper. Emphasis is placed on data
relevant to the observations and modelling of wind-induced snow transport in alpine terrain. This process strongly influences
the spatial distribution of snow cover in mountainous terrain with consequences for snowpack, hydrological and avalanche
hazard forecasting. In-situ data consist of wind (speed and direction), snow depth, and air temperature measurements
(recorded at four automatic weather stations), a database of blowing snow occurrence and measurements of blowing snow
fluxes obtained from a vertical profile of Snow Particle Counters. Observations data span the period from December 1st to
March 31st for each winter season from 2000-2001 until 2015-2016. The time resolution varies from 15 min. at the beginning
of the period to 10 min. for the last years. Atmospheric data from a local meteorological reanalysis (SAFRAN) are also
provided from 1 August 2000 to 1 August 2016. A Digital Elevation Model (DEM) of the study area (1,5 km²) at 20-cm
resolution is also provided in RGF 93 Lambert 93 coordinates  This dataset has been used in the past to develop and evaluate
physical parameterizations and numerical models of blowing and drifting snow in alpine terrain. Col du Lac Blanc is also a
target site to evaluate meteorological and climate models in alpine terrain. It belongs to the Cryobs-Clim observatory (the
CRYosphere, an OBServatory of the CLIMate) which is a part of the national research infrastructure OZCAR (Critical Zone
Observatories – Application and Research) (Gaillardet et al., 2018). The data are placed on the repository of the OSUG
datacenter doi:10.17178/CRYOBSCLIM.CLB.all .





## 1 Introduction


Wind-induced snow transport strongly influences the temporal and spatial distribution of the snow cover in mountainous areas (e.g. Mott et al, 2010, Vionnet et al. 2014). It occurs throughout the winter in a succession of blowing snow events with and without concurrent snowfall. The redistribution of snow through saltation and turbulent suspension results from complex interactions between the local topography, the near-surface meteorological conditions and the surface of the snowpack (e.g.
Pomeroy and Gray, 1995, Naaim-Bouvet et al., 2010). This spatial variability has consequences on the snowpack stability and influences the danger of avalanches as cornices and wind slabs are formed during blowing snow events. It has also hydrological consequences since the melt response of alpine catchment depends on the snow spatial distribution at peak accumulation (Egli et al., 2012, Revuelto et al., 2016). Observations of blowing snow and associated meteorological and snowpack parameters are therefore crucial to better understand the complex snowpack/atmosphere interactions during
blowing snow events and to develop and evaluate numerical models used in support of avalanche hazard and hydrological forecasting in alpine terrain.

Since the beginning of the 90', the Snow Research Centre (Météo-France – CNRS) and the ETNA unit (IRSTEA, Univ. Grenoble Alpes) have joined their efforts to investigate in-situ the effects of wind on snowpack evolution, during or after snowfall. A high-altitude experimental site has been set up at the Col du Lac Blanc, a north-south oriented pass, located at
2720 m altitude (45.13°N, 6.12°E) in the Grandes Rousses mountain range, France. Recent studies have focused on fine scale processes during blowing snow events (Naaim Bouvet et al., 2010, 2011, 2013; Nishimura et al., 2014; Schön et al., 2015), intercomparaison of blowing snow sensors (Cierco et al., 2007; Trouvilliez et al., 2015) and the development and evaluation of blowing snow models (Durand et al., 2005; Vionnet et al., 2013, 2014, 2017, 2018). In this paper, we present a unique meteorological and blowing snow dataset for each winter of the period 2000–2016. Meteorological data are available
from 4 automatic weather stations (AWS) surrounding the experimental site. Blowing snow data stem from two sources: (i) a database of blowing snow occurrence and (ii) blowing snow fluxes derived from Snow Particles Counters over the last 6 years. The paper is organized as follows. Section 2 describes the experimental site and the sensors used at each automatic weather station. The methods applied to derive blowing snow data are also mentioned. Then, Section 3 presents an overview of the meteorological and blowing snow dataset over the last seasons. Finally, Section 4 details the data availability.




## 2 Data description

### 2.1 Site description

The Col du Lac Blanc (CLB) experimental site, located at 2720 m altitude in the Grandes Rousses mountain range (45.13°N, 6.12°E, Fig. 1), has been operated by Météo France and Irstea with punctual collaboration with other academic partners, since 1988. This experimental site can be assimilated to a natural wind tunnel due to its orientation and the specific configuration of the surrounding summits. Indeed, the Grandes Rousses range on the eastern side and the "Dôme des Petites Rousses" summit on the western side channel the atmospheric flow according to a North-South axis (Fig 1). This

characteristic of the site is particularly useful for studies on the effects of wind on snow redistribution. Snow is typically present on the ground around the site (Fig. 2) from late October to early June with a strong inter-annual variability. The underlying ground is covered by bare and rocky soil, typical of high altitude alpine regions. Patches of low alpine grass are also present around the site. In wintertime, the site is characterized by a strong spatial variability of the snowpack due to intense snow redistribution during blowing snow events. In particular, two 10-m slope breaks on the northern and southern

side of the pass accumulate large amount of snow during these events depending on the main wind direction (Vionnet et al., 2014, Schön et al, 2015, 2018).

The CLB experimental site consists of four automatic stations located around the pass (Fig. 1 and 2). The exact position and elevation of the stations is given in Table 1. AWS Lac Blanc, Col and Muzelle are located at approximately the same elevation next to the pass where the atmospheric flow is strongly channeled by the surrounding topography. AWS Dome lies

at higher elevation on the top of the Dôme des Petites Rousses. At this station, atmospheric conditions are less influenced by the topography of the pass and closer to the synoptic conditions. Two wooden shelters are also installed at the pass (Fig. 2). They host the data acquisition system, the equipment storage and the living facilities. They are located on the eastern side of the pass, aside from the main wind direction, so that they have a minimum impact on measured snow and meteorological conditions at AWS Lac Blanc, Col and Muzelle. The meteorological, snow depth and blowing snow data collected at these

stations are presented in the next two sections.





**Figure 1: Location of the Col du Lac Blanc experimental site seen at different scales. Map (c) shows the location of the four AWS surrounding the site and described in Table 1.**



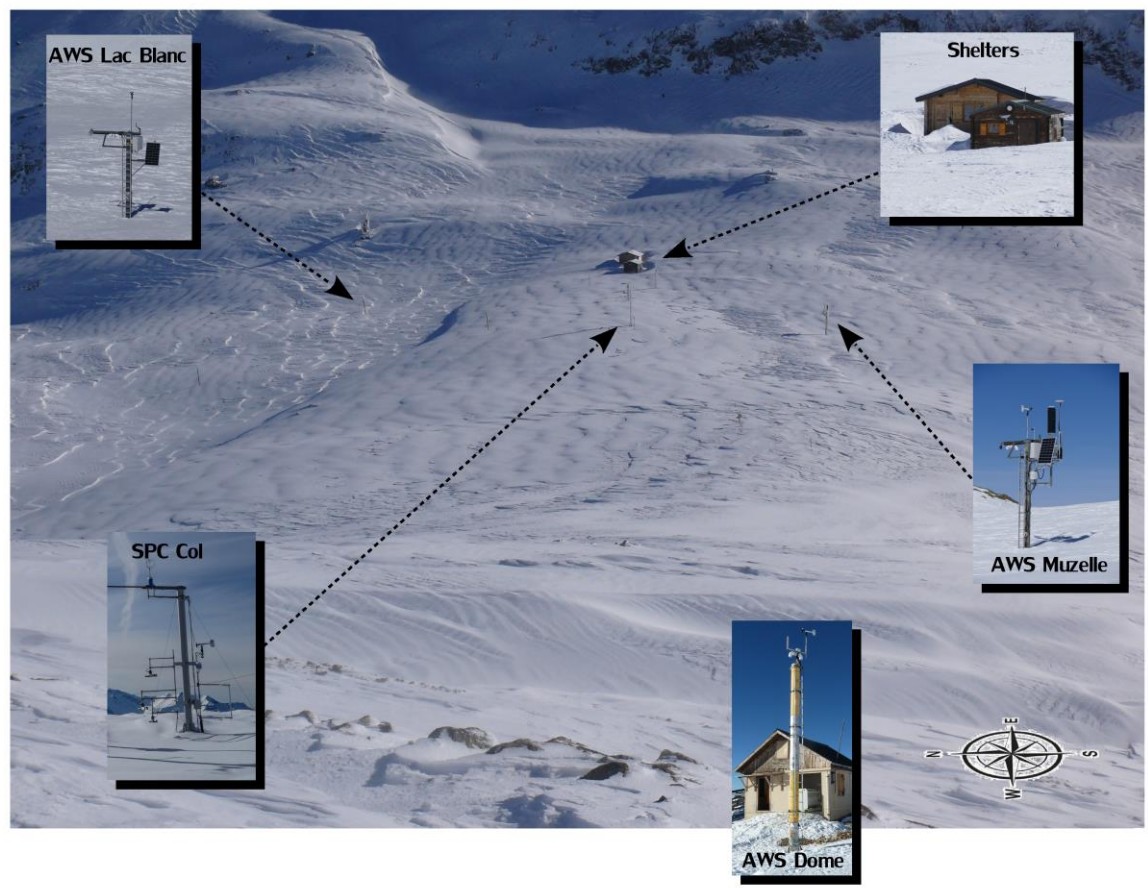

**Figure 2: Overview of the experimental site at Col du Lac Blanc (2720 m altitude, Grandes Rousses mountain range, France). Insets show the detailed view of each AWS. AWS Dome lies outside the picture. See text for further details on each AWS.**

**Table 1: List of the automatic stations at the Col du Lac Blanc experimental site.**

| Name | Latitude | Longitude | Altitude (m) | Measured data |
|---|---|---|---|---|
| Lac Blanc | 45° 7' 41,99" N | 6° 6' 43,36" E | 2710 | Air temperature, wind speed and direction, snow depth |
| Muzelle | 45° 7' 38,22" N | 6° 6' 40,11" E | 2721 | Air temperature, wind speed and direction, snow depth |
| Dome | 45°7'40.50" N | 6°6'20.94" E | 2806 | Air temperature, wind speed and direction. |
| Col | 45°7'39.19" N | 6°6'41.92" E | 2726 | Blowing snow flux, snow depth and wind speed and direction |



**Table 2: Overview of the sensors used at each station between 2000 and 2016 at Col du Lac Blanc, France.**

| Station | Variable | Sensors | Period of operation | Height* | Units | Integration method |
|---|---|---|---|---|---|---|
| AWS Lac Blanc | Air temperature | PT 100/4 wires | 12/2000→07/2014 | 6 m | K | Integrated (15 min) |
|  |  |  | 07/2014 → … | 6 m | K | Integrated (10 min) |
| AWS Lac Blanc | Wind speed & direction | Young 05103 | 12/2000→07/2014 | 7.5 m | m.s$^{-1}$ | Integrated (15 min) |
|  |  |  | 07/2014 → … | 7.5 m | m.s$^{-1}$ | Integrated (10 min) |
| AWS Lac Blanc | Snow depth | Cimel | 12/2000→07/2014 | 6.15 m | m | Integrated (15 min) |
|  |  |  | 07/2014 → … | 6.15 m | m | Integrated (10 min) |
| AWS Muzelle | Air temperature | PT 100/4 wires | 12/2000 →07/2014 | 5.5 m | K | Integrated (15 min) |
|  |  |  | 07/2014 →… | 5.5 m | K | Integrated (10 min) |
| AWS Muzelle | Wind speed & direction | Young 05103 | 12/2000→07/2014 | 7 m | m.s$^{-1}$ | Integrated (15 min) |
|  |  |  | 07/2014 →… | 7 m | m.s$^{-1}$ | Integrated (10 min) |
| AWS Muzelle | Snow depth | Cimel | 12/2000→09/2012 | 5.75 m | m | Integrated (15 min) |
|  |  | Cambpell Scientific SR50A | 09/2012 → 07/2014 | 5.75 m | m | Integrated (15 min) |
|  |  |  | 07/2014 →… | 5.75 m | m | Integrated (10 min) |
| AWS Dome | Air temperature | PT 100/4 wires | 12/2000→ 07/2014 | 7.5 m | K | Integrated (15 min) |
|  |  |  | 07/2014 → … | 7.5 m | K | Integrated (10 min) |
| AWS Dome | Wind speed & direction | Young 05103 | 12/2000→ 07/2014 | 8.5 m | m.s$^{-1}$ | Integrated (15 min) |
|  |  |  | 07/2014 → | 8.5 m | m.s$^{-1}$ | Integrated (10 min) |
| AWS Col | Blowing snow flux | SPC | 01/2011 → ... | 0.2→2.5m | g.m$^{-2}$.s | Integrated (10 min) |
| AWS Col | Wind speed & direction | USA1 | 01/2011 → 03/2011 | 5.7 m | m.s$^{-1}$ | Integrated (10 min) |
|  |  | Young 05103 | 12/2011 → 03/2012 | 3.7 m |  |  |
|  |  | USA1 | 12/2012 → 03/2013 | 5.7 m |  |  |
|  |  | Young 05103 | 12/2013 → 03/2014 | 3.7 m |  |  |
|  |  | USA1 | 12/2014 → ... | 5.7 m |  |  |
| AWS Col | Snow depth | SHM30 | 12/2010 → ... | 4.8m | m | Integrated (10 min) |

*Height above snow-free ground

## 2.2 Meteorological and snow depth data

Table 2 provides an overview of the meteorological and snow parameters measured around CLB, with the corresponding instrument type and height. Each AWS located around CLB measures wind speed and direction at a time step of 15 min or 10 min depending on the station and/or the period. Additionally, measurements of air temperature are available at three AWS. Finally, three AWS situated around the pass measure snow depth using ultrasonic sensors (at AWS Lac Blanc and Muzelle) and laser sensor (at AWS Col). These data have undergone a careful manual quality check. They are available between 1$^{st}$ December and 31$^{st}$ March of each winter. This period has been selected since it corresponds to the main period during which most of blowing snow events occur at CLB (Vionnet et al., 2013). Data from AWS Lac Blanc and Dome are available from winter 2000-2001 to 2015-2016 whereas data from AWS Muzelle and Col are available from winter 2002-2003 and 2010-2011, respectively. In complement to in situ meteorological observations, atmospheric data from the SAFRAN meteorological analysis system (Durand et al. 1993) are provided for Col du Lac Blanc from August 1$^{st}$ 2000 to July 31$^{st}$ 2016.



### 2.2.1 Wind speed and direction

Wind speed and direction are measured using non-heated anemometers (Young) at AWS Lac Blanc, Muzelle and Dome. The starting threshold of the wind velocity for the Young sensor is 1 m.s$^{-1}$. The maximum, minimum and mean wind velocity of
the time step are recorded. At AWS Col, wind speed and direction are measured using a heated ultrasonic anemometer from Metek Corporation (USA1). For high drifting snow fluxes, particles hitting the transmission/reception cell can disturb the measurement process The recorded wind direction is the most frequent for the time step. Due to snow accumulation at the bottom of the stations, the measurement height is changing during the course of the winter. Table 2 gives the height of the wind sensors over snow-free ground for each station. Snow depth measurements at AWS Lac Blanc, Muzelle and Col (see
Sect. 2.2.3) can be used to retrieve the height of the wind sensor above the snow surface as in Vionnet et al. (2013).

### 2.2.2 Air temperature

Air temperature is measured with PT100 wires at AWS Lac Blanc, Muzelle and Dome. The sensors are placed in ventilated shelter and the uncertainty in the measurements lies within 0.1 K.  Due to snow accumulation at the bottom of the stations, the measurement height is changing during the course of the winter. Table 2 gives the height of the temperature sensors over
snow-free ground for each station.

### 2.2.3 Snow depth

Snow depth is measured using ultra-sound depth sensors at AWS Lac Blanc and Muzelle. The correction of the impact of air temperature on the velocity of sound in the atmosphere is carried out using the air temperature measurement previously described. The data are further manually corrected to remove outliers in the dataset, most often occurring during snowfall.
Ultra-sound depth sensors provide measurements accurate within 1 cm for a surface area of a few cm$^2$ on the ground. The overall accuracy of the automated snow depth record is thus on the order of 1 cm.
Snow depth is measured using a laser sensor (SHM30) at AWS Col. The sensor uses an optoelectronic distance measurement principle to achieve a specified measurement uncertainty of better than 5 mm. Divergence of the SHM30 laser beam amounts to 0.6 mrad, which implies that the beam diameter is up 11 mm in size at the measurement point. This sensor is
mainly used to accurately determine the position of Snow Particle Counters above the snow layer during a blowing snow event. Moreover, this technology has the advantage of being less disturbed by blowing snow particles than the sonic sensor but its power consumption is higher.

### 2.2.4 Atmospheric parameters from a meteorological reanalysis

Atmospheric data from the SAFRAN meteorological analysis system (Durand et al. 1993) at the elevation of Col du Lac
Blanc in the Grandes Rousses range are provided from August 1$^{st}$ 2000 to July 31$^{st}$ 2016. SAFRAN combines meteorological fields from the numerical weather prediction system ARPEGE with neighboring observation to get an estimation of



meteorological parameters in the French mountains. It is used operationally in support of avalanche hazard forecasting (Lafaysse et al., 2013). SAFRAN data at CLB are provided to get all the meteorological parameters required to run continuously a land surface model at CLB without the need to restart the system every winter. SAFRAN data includes 2-m
wind speed, 2-m air temperature and humidity, incoming longwave and shortwave radiation and snowfall and rainfall amount at an hourly time step. In particular, SAFRAN provides a valuable estimation of the snowfall amount at Col du Lac Blanc that cannot be precisely measured at the site during snowfall under windy conditions due to strong gauge undercatch. Therefore, SAFRAN precipitation is considered as the reference precipitation dataset at Col du Lac Blanc. On the other hand, SAFRAN tends to underestimate wind speed at CLB due to the influence of the surrounding topography which is not
included in the conceptual representation of the topography in SAFRAN. It is recommended to replace SAFRAN wind speed and direction by the observations collected at CLB when running a land surface scheme at CLB in wintertime as described in Vionnet et al. (2013).

## 2.3 Blowing snow data

Blowing snow data stem from two sources: (i) a database of blowing snow occurrence from winter 2000-2001 to winter 2015-2016 and (ii) blowing snow fluxes derived from Snow Particles Counters from winter 2010-2011 to winter 2015-2016. These two datasets are described below as well as comparison of the occurrence of blowing snow with the two methods.

### 2.3.1 Database of blowing occurrence

A database of blowing snow events at CLB was established for each winter of the period 2000-2016. It consists in an
extension of the database presented in Vionnet et al. (2013). A blowing snow event is defined as a time-period when snow on the ground was transported in the atmosphere in saltation and in turbulent suspension. Such event may occur with concurrent snowfall. Therefore, the database makes the distinction between blowing snow events with and without concurrent snowfall. The identification of a blowing snow event in the database relied on an empirical method that required the combination of several datasets (wind, precipitation and snow depth) and their detailed analysis by an expert. This
method is extensively described in Vionnet et al (2013) and its main characteristics are summarized here.

Periods of concurrent snowfall and ground snow transport were identified first. They are defined as periods with precipitating snow (total snowfall from SAFRAN reanalysis greater than 5 $mm_{SWE}$ over the period) with a 5 m-wind speed above the snow surface, $U_5$, at AWS Lac Blanc higher than $U_{5t} = 6$ m.s$^{-1}$. This threshold wind speed followed the observations of Sato et al (2008) collected during their wind-tunnel experiments on the processes of fracture and
accumulation of snowflakes. $U_5$ is obtained from wind speed and snow depth measured at AWS Lac Blanc using a standard log-law for the vertical profile of wind speed near the surface, as in Vionnet et al. (2013). Periods of ground snow transport were then identified at an hourly time step from an analysis of the recordings from the snow depth sensor. This indirect method was developed and tested over fifteen years of observations at Col du Lac Blanc and has been previously used in



Guyomarc'h and Mérindol (1998) and Vionnet et al (2013) to identify periods of snow transport. Positive values of the
difference between the maximum and minimum snow depth recorded over an hour and associated large values of its standard
deviation are characteristic of the presence of snow particles between the sensor and the surface of the snowpack. Snow
particles in motion above the snowpack surface create indeed interference in the ultrasonic signal. The records from a
webcam (installed in 2004) completed the analysis. Such an empirical method provides the start and end dates of each
blowing snow event. Only events of duration longer than 4 h were recorded in the database. Overall, the database contains at
an hourly time step information on the occurrence of blowing snow and the type of event (with or without snowfall).

Figure 3 shows an overview of the inter-annual variability of blowing snow occurrence reported in the database with a
winter average ranging from 6.4% in 2002-2003 to 19.0 % in 2011/2012. Using this method, blowing snow occurred during
11.7 % of time at Col du Lac Blanc over the period 2000-2016. 36.7 % of time blowing snow occurred with concurrent snow
fall. These estimations are similar to those reported in Vionnet et al. (2013) for the period 2001-2011.

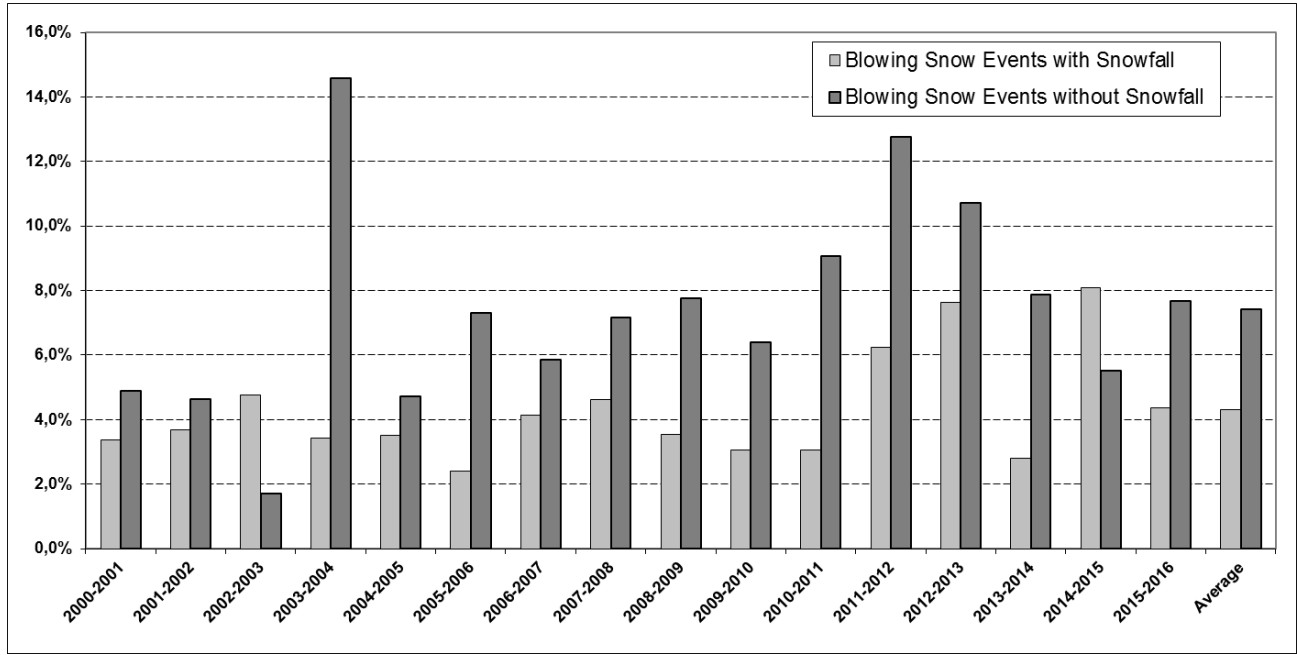


**Figure 3: Percentage of time when blowing snow events (with and without snowfall) are recorded for each winter (December 1st to March 31st) over the period 2000-2016 at Col du Lac Blanc experimental site.**

**2.3.2 Data from Snow Particle Counter**

The Snow Particle Counter (SPC-S7, Niigata Electric) is an optical device (Sugiuria et al., 1998) detecting snow particles
between 40 and 500 µm in mean diameter by their shadows on photodiode. The SPC-S7 has a self-steering wind vane and
the sampling area, perpendicular to horizontal wind vector, is 50 mm$^2$ (2 mm × 25 mm). Assuming spherical snow particles,
the horizontal snow mass flux can be calculated as follow:

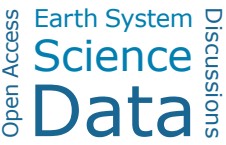


$$q = \sum q_d = \frac{\sum n_d \frac{4}{3}\pi \left(\frac{d}{2}\right)^3 \rho_p}{S * t}$$

Where $q_d$ is the horizontal snow mass flux for the diameter d, $n_d$ is the number of drifting snow particles, S the sample area, t
the sample period, and $\rho_p$ the density of the drifting snow particles (917 kg m$^{-3}$).

Depending on the winter season, up to four Snow Particle Counters SPC-S7s were installed on a mast (Bellot et al., 2016).
The SPC-S7s could be raised manually when the snow depth increased and risked burying the sensors. Horizontal snow flux
is highly dependent on height above the snow surface. Because of snowfall and blowing snow events the elevation of SPC
above the snow varied substantially during the winter season preventing any direct comparison over time being made. That's
why in the present database, the snow fluxes near the surface (that means the snow flux provided by the SPC closest to the
snow surface) and the corresponding height of the sensor are provided. When available, snow fluxes at a higher position are
used to standardize the horizontal snow flux and to estimate the mean horizontal snow flux at 1 m above the snow surface
and vertically integrated over 1 m (between 0,2 and 1,2 m over the snow surface). The computation of these fluxes is
described below.

According to the diffusion theory of snowdrift, it is possible to approximate averaged drift density [kg m$^{-3}$] as a function of
height and wind velocity (Radok, 1977, Gordon et al. 2009). If the average wind profile is approximated by a power law, the
vertical distribution function for the snow flux $\mu$ (g.m$^{-2}$.s$^{-1}$) is expressed as follow:

$$\mu(z) = A \cdot z^{-m}$$

where A is a calibration parameter and m the exponent which is independent of z; both are derived by regression from
measured data (Trouvilliez et al., 2015, Bellot et al., 2016). From this, the mean horizontal snow flux at 1 m and vertically
integrated over 1 m can be estimated.

Nevertheless, this typical profile is not always observed or cannot be determined in situations when:

i)       only one sensor provides data

ii)      snow flux at different heights are not significantly different (case of falling snow with low wind,…)

iii)     data are physically inconsistent (flux at the higher position is much greater than near the surface due for example to
icing of the self-steering windvane).

Therefore the SPC data were processed as follows. First a filter was applied to the raw data to supress possible electronic
noise: events with a particles flux smaller than 20 particles cm$^{-2}$ during 10 minutes were discarded. Then, an algorithm was
designed and used to categorize each 10-min profile of SPC data set into six different groups (specified in the database):

• "Undetermined": only one sensor provides data making it impossible to do the vertical interpolation.

   • "Power_law": the vertical distribution function can be expressed as a power law.

   • "Inconsistent": Data are physically inconsistent.





- "Mean": snow fluxes at different heights are not significantly different, so that the mean horizontal snow flux at 1 m is estimated by the average of the measured flux.

- "No_flux": In this case no sensor of the SPC vertical profile detects more than 20 particles per cm² during ten minutes.

- "Maintenance": During sensor maintenance, data acquisition is stopped.

When possible ("Power law" and "Mean"), mean horizontal snow flux at 1 m and vertically integrated over 1 m (between 0,2 and 1,2 m) are determined.

Figure 4 shows an overview of the inter-annual variability of blowing snow occurrence and intensity derived from the SPC data. SPC provided a non-zero value during a percentage of time ranging from 43 % in 2010-2011 to 63 % in 2013/2014 corresponding to an average quantity of snow transported between 0.2 and 1.2 m per linear meter during 30 days ranging from 2547 kg in 2010-2011 to 10331 kg in 2013/2014. SPC provided a non-zero-value during 50% of time and an average quantity of snow transported between 0.2 and 1.2 m per linear meter and per 30 days of 6245 kg during the winter season
over the period 2010-2016. Such percentage is quite different from those reported in the empirical database (Fig. 3) but we have to keep in mind that SPC which detects each particle is able to identify trace precipitation which do not significantly contribute to blowing snow quantities. This point is discussed in details in the next paragraph.

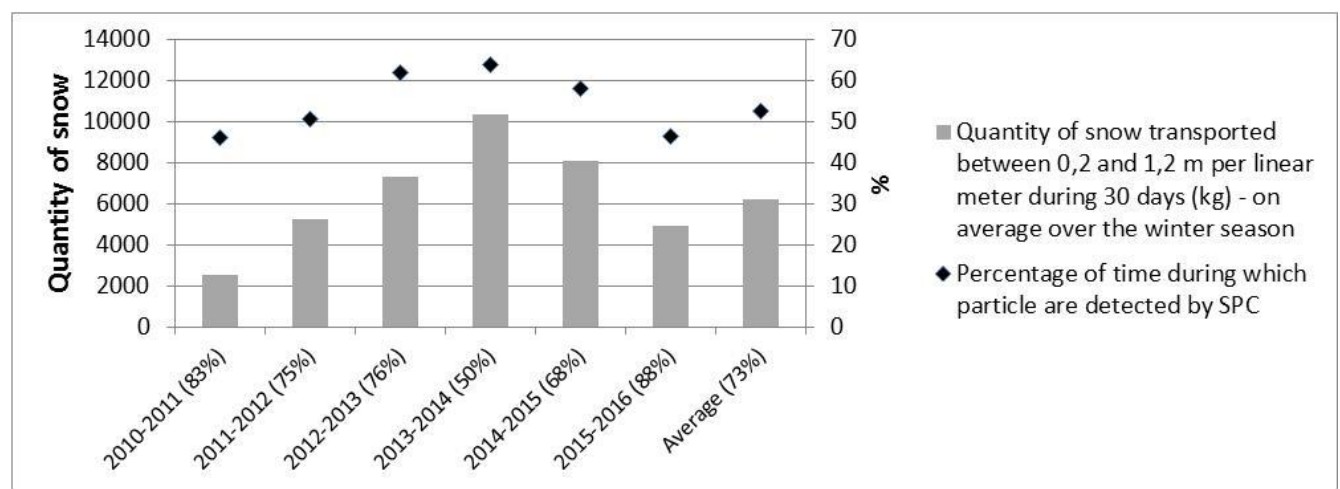

**Figure 4: Percentage of time when particles are detected for each winter (December 1st to March 31st) over the period 2010-2016 at**
**Col du Lac Blanc experimental site and the corresponding quantity of snow transported between 0.2 and 1.2 m per linear meter. Due to missing or invalid SPC data, the length of the time series varies from a winter season to another winter season preventing to easily study the inter-annual variability of blowing snow intensity. To overcome this, an average quantity of snow transported between 0.2 and 1.2 m per linear meter during 30 days has been calculated. Numbers in brackets indicates the percentage of valid data delivered by SPC during the considered winter season.**




### 2.3.3 Comparison of the two methods used for the determination of blowing snow periods

During wintertime, SPC provided a non-zero-value during 50 % of time (and an average quantity of snow transported between 0.2 and 1.2 m per linear meter and per 30 days of 6245 kg during the winter season over the period 2010-2016) whereas blowing snow occurrence reported in the empirical database is 12% during the same period. This call for detailed

comments.

Periods without significant blowing snow fluxes should be excluded to take the analysis a few steps further. That's why a filter has been applied over the raw SPC data: events with a particles flux smaller than 120 particles $cm^{-2}$ $min^{-1}$ (Naaim-Bouvet et al., 2014) were considered as periods without drifting snow. Then similar data processing was applied with detection threshold parameters of 960 particles.$cm^{-2}$.$min^{-1}$ to highlight the global trends. Table 3 compares the occurrences

of blowing snow derived from the SPC data with the different threshold to the occurrences reported in the empirical database.

**Table 3: Comparison between observed (SPC) and estimated (empirical method) drifting snow events in terms of occurrence and quantity.**


| | | | 2010-2011 | 2011-2012 | 2012-2013 | 2013-2014 | 2014-2015 | 2015-2016 | Average |
|---|---|---|---|---|---|---|---|---|---|
| Occurrence of drifting snow (Snow Particle Counter) | | Threshold (2p/cm²/min.) | 43% | 48% | 63% | 62% | 59% | 31% | 50 |
| | | Threshold (120p/cm²/min.) | 30 | 36 | 44 | 41 | 39 | 30 | 36 |
| | | Threshold (960p/cm²/min.) | 19 | 24 | 27 | 25 | 22 | 19 | 23 |
| Total quantity of snow transported between 0,2 and 1,2 m per linear meter during the winter season (kg/m) estimated from SPC | | Threshold (2p/cm²/min.) | 6290 | 15989 | 22288 | 20913 | 22248 | 17567 | 6245 kg/30 days |
| | | Threshold (120p/cm²/min.) | 6249 | 15904 | 22149 | 20840 | 22133 | 17543 | 6073 kg/30 days |
| | | Threshold (960p/cm²/min.) | 6018 | 15368 | 21382 | 20450 | 21417 | 16796 | 5703 kg/30 days |



| | | 2010-2011 | 2011-2012 | 2012-2013 | 2013-2014 | 2014-2015 | 2015-2016 | Average |
|---|---|---|---|---|---|---|---|---|
| Occurrence of drifting snow (Empirical method see 2.3.1) | | 9% | 16% | 17% | 11% | 8% | 10% | 12 |
| % Total quantity of snow transported between 0,2 and 1,2 m per linear meter recorded during period of drifting snow identified by the empirical method see 2.3.1) | Threshold (2p/cm²/min.) | 62 | 66 | 52 | 65 | 32 | 66 | 55 |
| | Threshold (120p/cm²/min.) | 62 | 66 | 52 | 65 | 33 | 66 | 55 |
| | Threshold (960p/cm²/min.) | 64 | 68 | 53 | 66 | 33 | 69 | 56 |

When looking at the Figure 4 and Table 3, it can be concluded that :

-      The empirical method is quite robust compared to SPC measurements: on average over the six years, a quarter of the SPC measurements can be considered as invalid or are missing whereas continuous data are available in the database of blowing occurrence

-      The empirical method misses low to moderate blowing snow events (only 12% of time are considered as drifting snow event mainly due to the assumptions used in the method (events lasting longer than 4 h, wind speed greater than 6 m s$^{-1}$ for events with concurrent snowfall). In fact, occurrence of wind-induced snow transport (including low windy snowfalls) is closer to 30% of the time at Col du Lac Blanc. The exact value of percentage depends on the chosen threshold value when filtering the SPC data.

-      The empirical method is able to capture the main drifting snow events in terms of intensity (56%)




## 3 Overview of the last seasons

Figure 5 and 6 illustrate the meteorological and blowing snow conditions at the site over the last six snow seasons (2010-
2016). In particular, Figure 5 shows the strong control exerted by the local topography on the atmospheric flow at the pass.
Indeed, the wind field at AWS Lac Blanc (Fig. 5a) is characterized by a channelling along a North-South axis and an
increase in wind speed compared to AWS Dome (Fig 5c), located on the top of the Dôme des Petites Rousses and less
influenced by the local topography (Fig. 1). The distribution of blowing snow fluxes (Fig. 5b) is quite consistent with the
distribution of wind speed at the site (Fig. 5a). Figure 6 shows an overview of snow depth, maximum wind speed and
blowing snow fluxes measured at the site over the last six seasons. A strong inter-annual variability is found in terms of
snow depth with a maximum snow depth reaching 4.01 m in 2012/2013 and only 2.84 m for winter 2013/2014. Events with a
maximum wind speed above 15 m s$^{-1}$ are frequently measured at the site. These windy events are generally associated with
blowing snow events if erodible snow is present at the snow surface. The intensity of these events varies greatly (note the
logarithmic scale on the graphs showing blowing snow fluxes). Additional and more detailed summary plots for each year of
the present dataset are provided as Supplement to this article.

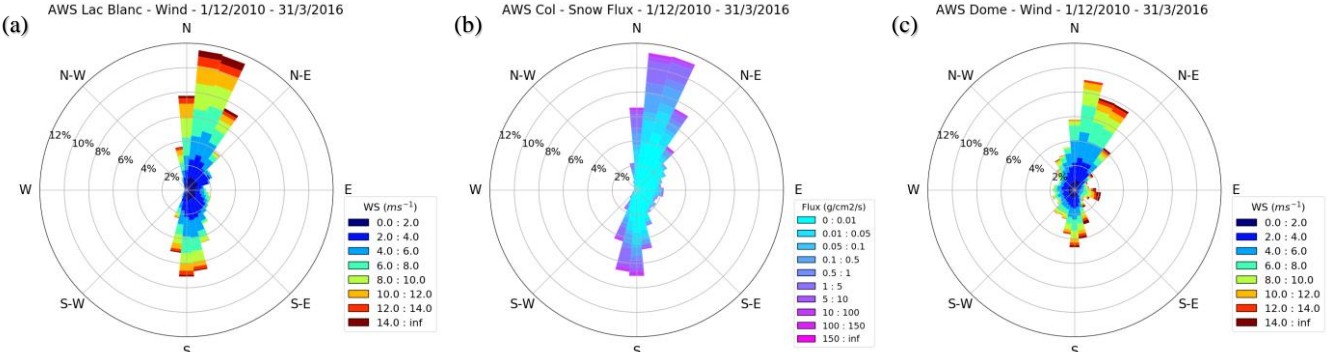

**Figure 5: Overview of the period 2010-2016: (a) wind rose for the AWS Lac Blanc, (b) rose of blowing snow fluxes at AWS Col
with snow flux vertically integrated over 1 m between 0.2 and 1.2 m and (c) wind rose for the AWS Dome. Yearly wind roses and
roses of blowing snow fluxes are provided as Supplement online material.**





**Figure 6 : Overview of snow depth (AWS Lac Blanc), maximum wind speed (AWS Lac Blanc) and blowing snow fluxes measured by SPC vertically integrated between 0.2 and 1.2 m (AWS Col) from 2010 to 2016. More detailed yearly graphs are provided as Supplement online material for each snow season from 2000-2001 to 2015-2016.**



## 4 Data availability

The database ([doi:10.17178/CRYOBSCLIM.CLB.all](doi:10.17178/CRYOBSCLIM.CLB.all)) presented and described in this article is available for download at http://doi.osug.fr/public/CRYOBSCLIM_CLB/index.html. Data at the different stations
([doi:10.17178/CRYOBSCLIM.CLB.COL](doi:10.17178/CRYOBSCLIM.CLB.COL), [doi:10.17178/CRYOBSCLIM.CLB.DOME](doi:10.17178/CRYOBSCLIM.CLB.DOME), [doi:10.17178/CRYOBSCLIM.CLB.LACBLANC](doi:10.17178/CRYOBSCLIM.CLB.LACBLANC), [doi:10.17178/CRYOBSCLIM.CLB.MUZELLE](doi:10.17178/CRYOBSCLIM.CLB.MUZELLE), [doi:10.17178/CRYOBSCLIM.CLB.SAFRAN](doi:10.17178/CRYOBSCLIM.CLB.SAFRAN)) from the Blowing Snow Occurrence Database ([doi:10.17178/CRYOBSCLIM.CLB.BSO](doi:10.17178/CRYOBSCLIM.CLB.BSO)) and from the SAFRAN ([doi:10.17178/CRYOBSCLIM.CLB.SAFRAN](doi:10.17178/CRYOBSCLIM.CLB.SAFRAN)) reanalysis are given in separated files in csv format. The Digital Elevation Model of the study area (1,5 km² -see
Supplementary Materials) at 20-cm resolution is also provided in RGF 93 Lambert 93 coordinates ([doi:10.17178/CRYOBSCLIM.CLB.DEM](doi:10.17178/CRYOBSCLIM.CLB.DEM)). Data of winters 2016-2017 and 2017-2018 and of the upcoming years will be added in the database on a yearly basis and made available to the community through the CRYOBSCLIM data portal (http://data.cryobsclim.fr).

## 5 Conclusions

The Col du Lac Blanc is a unique high-altitude experimental site in the French Alps where meteorological and blowing snow data are automatically acquired. This site is dedicated to studies of snow/atmosphere interactions and wind-induced snow transport. 16 years of quality controlled data have been combined and made freely accessible for the scientific community. They are composed of meteorological data from four automatic stations and a reanalysis product, a database of the main blowing snow events that occurred at the site over 16 years and 6 years of automatic measurements of blowing snow fluxes
with Snow Particles Counters. These high-quality data have been already used to evaluate snow redistribution models for specific blowing snow events or years (Schön et al., 2015; Vionnet et al., 2014, 2017, 2018) and to carry out detailed analysis of the physics of snow transport (Naaim Bouvet et al., 2010, 2011; Nishimura et al., 2014). We anticipate that these data will be used in the future in modelling studies of snow accumulation over multiple snow seasons and to better understand the links between the occurrence and intensity of blowing snow events and meteorological and snow conditions.
We hope that further instrumental developments will allow to improve the monitoring of snow/atmosphere interactions at this site for the benefits of the snow, avalanche, mountain hydrology and weather forecasting community.

## Acknowledgments

The authors would like to thank Fred Ousset, and Xavier Ravanat for their valuable help in the field. We also would like to acknowledge SATA (Sociètè d'Aménagement Touristique de l'Alpe d'Huez et des Grandes Rousses) which provided
logistical support and OSUG-DC for hosting the data. Col du Lac Blanc is a part of IR OZCAR and receive financial support from OSUG, Meteo France and Irstea. CNRM/CEN and Irstea are part of LabEx OSUG@2020 (ANR10 LABX56).



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
