# Peer review of "A meteorological and blowing snow dataset (2000-2016) from a high-altitude alpine site (Col du"

_Earth System Science Data, 2018_

## Referee Comment (RC1) · Anonymous Referee #1 · 18 Jul 2018

General comments:

This paper presents a meteorological database of blowing snow events for the Col du Lac Blanc study site a high-altitude experimental site located in Grandes Rousses range (French Alps). In-situ observations are obtained in four different automatic weather stations located within the study site. Additionally meteorological information is completed with SAFRAN model reanalysis. It is also described the methodology for obtaining blowing snow events and the data obtained with this methods are included in the database. The dataset described in this article has a great potential for many applications for studying snow dynamics on mountain areas. For this reason the manuscript

should be published. Nevertheless there are some issues that must be addressed before its final publication.

Major points:

1.- I encourage to include a new section explaining and describing Blowing snow data and the methods used (this is, change section 2.3 to section 3), since this probably is the most novel part of the paper. This section must clearly state from which AWS are the data. Sometimes it is difficult to follow this. Moreover Table 3 must be divided. You first present results obtained in section 2.3.2 with the particles thresholds. Afterwards you present a new "Table 4" with results shown in the two last rows of Table 3 since you are using there same method to compare the occurrence of drifting snow. This will help to understand the table faster for potential readers. I also miss some discussion about the fact that in Figure 4, when less data are available (percentage of valid data derived from SPC) more quantity of snow is detected and a higher percentage of time detected particles.

2.- I have missed some information about the climatic characteristics of the study site. As authors say, the experimental site has been operationally used since 1988. I think it is really interesting to provide an overview of the climatology observed in this site. For instance it could be included the mean annual and winter temperature, number of days with snow presence in the automatic weather station with the longest dataset, total annual precipitation. . .

3.- If possible, I encourage manuscript authors to include in the database observations obtained during the whole study period and not only during winter period. This can be really interesting since can provide an evaluation of observations/model deviations on an annual time basis. Moreover I think that observations of the last two snow seasons 2016-2017 and 2017-2018 are quite valuable, so I encourage manuscript authors to, upload this information during the review process.

4.- The is mostly focused on data obtained with different environmental sensors. This
way along the manuscript the model and the company (including a reference to their data) of sensors must be specified.

Specific comments:

Line 15: Precise that "Grandes Rouses" is located in French Alps.

Line 20: Precise the period in which the Snow Particle Counter acquired observations (2010-2016).

Line 28: Remove Gaillardet et al., 2018 reference. It is not appropriate to include a reference of an article submitted, even more if it is included in the abstract.

Line 43: Maybe rephrase as: "...have joined their efforts to investigate the effect of wind transport on snowpack evolution."

Line 44: "A high-altitude experimental site WAS set up..."

Line 45: By inspection of Figure 2. I guess that the study area covers an altitudinal range of about 200-300m. Please include maximum and minimum elevations in the text.

Line 53-55: Change appropriately in regard to the major comment of including a new section for describing blowing snow data.

Line 60: Describe the locations of Grandes Rouses within the Alps and include the altitudinal range of the study site.

Line 73: In table 1 and line 61, you provide the location of the automatic weather station on longitude, latitude; could you please also provide these coordinates on same coordinate system of the DEM available in the database?

Line 130: Which is the "manual quality check" process? You remove outliers?

Line138: Which young sensor? There are several products of this company.

Line 142: There is a final "." missed.

Line 147: Include the company of PT100 wires.

Line 149: You have already said that height changes during the course of the winter. Additionally you don't provide the snow free-height of these sensors. Remove this sentence.

Line 152: Please, include model and company of the ultra-sound snow height sensor. I also suggest giving a small explanation about how these sensors work.

Line 155: The surface area of the ultra-sound sensor observation may variate depending on snow height. Please clarify and quantify maximum and minimum surface area values.

Line 157: Include the company of SHM30 sensor.

Line 164: In the abstract you said that you provided SAFRAN reanalysis and here it is said that you provide SAFRAN analysis. Please clarify.

Lines 164-176: I see quite interesting to include SAFRAN model outputs. If I am right, this is not a 2D model. Please explain how you obtain the data for Col du Lac Blanc.

Line 173: You say SAFRAN is considered as the reference precipitation in Col du Lac Blanc; however this is a model and could have errors. Please discuss this issue and provide an estimation of potential bias of this model (even if it is for a different study area) in the Alps.

Line 195-204: These sentences are difficult to follow, please rephrase. For instance when you say: "Positive values of the difference...." I think you are describing the method you refer before as "This indirect method..." but this is not clear.

Line 203: How did you complete the analysis for the period 2000-2004 without the webcam? Maybe you could explain that the results obtained for the period 2004-2016 were evaluated with a visual inspection of webcam images.

Line 204: Change "recorded" by "included".

Line 225: Why 917 kg m-3 density value?

Line 228: Here you use the acronym SPC and not anymore SPC-S7s as you did before. Please be consistent along the manuscript when you refer to this device.

Line 240: Include A, z and m values you used to estimate mean horizontal flux and its vertical interpolation. Line 220 and 238: include a reference for mathematic equations (1, 2...).

Line 251: I guess this is the power law you introduce in line 238. Use a number to refer this expression.

Line 263 to 264: When you present "kg" of snow, specify that you are showing snow mass transport variable.

Line 266: "...to keep in mind that SPC, which detects each particle, is able to..."

Line 267: This is discussed in next section in several paragraphs...

Line 278: You already showed the 50% if time and the 6245 kg of mass on previous section. This is redundant. You can remove it here.

Line 291: I guess these conclusions came from table 3 since Figure 4 does not show the empirical method. Remove Figure 4 reference.

Line 299: I find quite surprising that the occurrence of wind-induced snow transport is closer to 30% of the time. Has been shown this value before? Where?

Line 301: You mean the empirical method with SPC data? Please clarify.

Line 316: Include mean snow depth value during the 2010-2016 time period.

Section 4, data availability: The database must include a metadata file for each AWS that includes all variables of each file, their units and the location of the station. Moreover I think it is not necessary to include the doi of each single station, for SAFRAN reanalysis and for the DEM all these links can be easily found following

the link: http://doi.osug.fr/public/CRYOBSCLIM_CLB/CRYOBSCLIM.CLB.all.html Concerning the DEM, it must be provided DEM on a single file and not in 14 separate files. The research group knows in detail the study site characteristics and any incoherence coming from alignment errors of the separate files can easily be detected, what it is not the case for potential users of the DEM. If necessary the spatial resolution of the DEM can be reduced to 0.5 m or 1 m grid cell size. X, Y and Z units and column names must be included in xyz files. Also a metadata diles of the DEM must be included.

Figures and tables:

Figure 1: Please include in c) panel the "Dôme des Petites Rouses" triangle and the point that marks "Col du lac Blanc" from b) map. I also encourage manuscript authors to draw dashed lines on c) map showing the area covered in picture of Figure 2. This would really help to potential readers to understand the characteristics of the study site.

Figure 5: I guess that the different circles of wind roses show the frequency of the different events. Please clarify. Maybe it is more interesting to provide the wind rose for AWS Col (same of (b) wind rose) since you are showing the fluxes obtained in this station.

Table 2: Please group first column when same station is described. It will be easier to understand the table.

Table 3: For the second column of the table, put "Threshold" in the first cell and not for each single cell. Units of the different variables are not appropriately included; in some cases the occurrence of drifting snow presents the % in others not. Similarly Kg of snow mass transport are not provided. Please be consistent along the table. Moreover there is a mistake and the 2p/cm2/min threshold is a 20p/cm2/min threshold as introduced in line 248. In the last row of table 3: You say that it is shown the "Total quantity of snow transported" however I think it is the occurrence of drifting snow. Moreover I see a bit confusing that you show in this row the results obtained with the method presented in section 2.3.1 with the SPC data without introducing this before.

Supplementary material: Line 20: Remove one "of" right before 500*500.

---

## Referee Comment (RC2) · Anonymous Referee #2 · 18 Jul 2018

The paper describes the meteorological and blowing snow dataset collected at Col du Lac Blanc (2720 m a.s.l.) in the French Alps. This data consists of wind speed and direction, air temperature, snow depth, blowing snow fluxes and occurrence periods, spanning the period from 2000/2001 until 2015/2016. The data is complemented with local atmospheric reanalysis from SAFRAN, and a digital terrain model with 20 cm resolution. All data are now available for the public, the dog's are given.

The paper is a very welcome contribution for the ESSD Special Issue: Hydrometeorological data from mountain and alpine research catchments, and for the scientific community of snow scientists. However, I recommend some improvements prior to

final publication.

General aspects:

- my major concern is that in the way it is presented, the comparison of the database of blowing snow occurrence and the SPC is not meaningful (2.3.1, 2.3.2 and 2.3.3). Both data sources make use of more or less empirical thresholds to classify a period as blowing snow occurrence, or to count the percentage of time during which particles are detected by the SPC, respectively. There is no better or worse of the two methods, they are only different and, due to the threshold values chosen, provide different results. Why do you choose thresholds in a way that the results become such different? The possibilities for improvement that I see here are (i) just present the two datasets "as is" without comparison, (ii) explicitly justify the choice of all thresholds, and explain the difference in the results, (iii) calibrate one method with the other, or (iv) leave the empirical database out und just provide a reference. In any case, always distinguish clearly physical processes from empirical estimates. Finally, it is not clear who the original author of the database methodology is, Guyomarc'h and Merindol (1998) or Vionnet et al. (2013)?

- my second major concern is a thorough discussion of the (very important) scale effects of blowing snow, and which ones are observed at Col du Lac Blanc. This should be an original paragraph in the introduction

- the fact that the data collection continues and new measurements will follow and be made available should be mentioned in the beginning of the paper, not at the end (in section 4)

- in figure 1 (the maps) the color scheme/contour lines should be improved: certain altitudes should be associated with a contour line, not with a color. The latter should be associated with an altitudinal range. In this map figure, the typeface and size should be harmonized. In the legend, "Automatic stations" should be "Automatic weather stations", or simply "AWS"

- Table 3 is more confusing than helpful; is there not better way of presenting these numbers?

- the reference section is full of type inconsistencies and mistakes. This entire section needs complete revision (should have been checked prior to submission)

- the use of lowercase and uppercase letters needs revision in the entire manuscript

Details:

- line 20: "Snow Particle Counter" in uppercase: why? Is this the name of the device? I would recommend "snow particle counter (SPC)", and use "SPC" only in the following text

- line 21/22: give the date when resolution changes

- line 28: avoid references in the abstract

- line 33: insert "atmospheric" between "concurrent" and "snowfall"

- line 51: "SPC"

- line 72: insert "weather" between "automatic" and "stations"

- line 90: better "a" instead of "the" ("detailed view")

- Table 1: remove the dot after "direction"

- Table 2: remove the dots in the units, insert space in "4.8m"

- line 133: the availability of the AWS Muzelle data given here does not match the one given in table 2

- line 1390: remove dot in "m.s-1"

- line 142: insert dot after "process"

- line 145: replace "Sect."

- line 162: better remove "but its power consumption is higher" - this is another issue of no relevance here

- entire section 2.2.4: make clear if the SAFRAN data presented here origins in an "analysis", or in a "re-analysis", and use the correct term then everywhere

- line 181: "SPC"

- line 183. insert "snow" between "blowing" and "occurrence"

- line 184: replace "of the period" with "in the period", and add at the end of the sentence where the sensor was established. Replace "consists in" with "consists of"

- line 186: replace "was" with "is", make "event" plural ("events")

- line 188: better "relies" instead of "relied", and "requires" instead of "required"

- line 189: replace "datasets" with "meteorological occurrences"

- line 191: better formulate "Periods of ground snow transport with concurrent snowfall are identified first"

- line 196: do you mean a logarithmic law? If yes, write the entire word and avoid abbreviations

- line 204: how did you choose the threshold of 4 hours? The results depend on it!

- line 207: does this correspond to the sum of the two columns in a winter season? Indicate this in the text. Avoid to write "blowing snow occurred" in the context of the empirical method with the threshold, since the process of blowing snow probably occurred much more often! (i.e., write something like "blowing snow periods were classified . . .")

- line 209: why only "are similar", should these estimations not be the same as in Vionnet et al. (2013)? What is different, and why?

- line 214: better write "Data from SPC devices"

- line 215: "SPC"

- line 216: insert "a" between "on" and "photodiode"

- line 216/217: better make two sentences out of the one

- line 226: "SPC"

- line 227: delete "and risked burying the sensors"

- line 228: better of "the" SPC

- line 229: delete "being made". Better write full words, i.e. "That is why"

- line 235: insert "the" between "approximate" and "averaged"

- line 338: correctly align the minus in the formula

- line 244: delete "case of", add "speed" (?) after "wind"

- line 248: delete the "s" at the end of "particles" (-> singular)

- line 258: better "the mean horizontal fluxes"

- line 261: "the" SPC data

- line 263: "The" SPC

- line 265: "in" the period

- line 266: "the" SPC

- line 266: . . . able to "identify trace" precipitation . . . ? Please clarify!

- Figure 4: add unit to y-axis and format typesetting properly. In the figure caption, better write "during" the period, and "delivered by the SPC". . .

- line 279: this "calls" for

- line 281: better "That is why"

- line 282: "applied for the raw data", with "a particle flux smaller . . ." (singular)

- line 283: "Then a similar data processing. . .", and: why "similar", and not "the same". Explain the difference, if it exists!

- line 284: "a" direction threshold. Delete "parameters"

- line 285: "with different thresholds"

- line 293: "to the SPC"

- line 295: "blowing snow occurrence"

- line 309: site "for the last six snow seasons"

- line 334: better "of" the different stations

- line 341: "The data of the winters. . ."

- line 350: "SPC"

- line 358: delete comma

- line 359: better "who" instead of "which"

Good luck, a very nice peace of work!

---

## Referee Comment (RC3) · Anonymous Referee #3 · 31 Jul 2018

General comment:

This manuscript presents wind, snow depth, and air temperature measurements from four automatic weather stations and measurements of blowing snow transport in a high-elevation site in the Col du Lac Blanc, France. This is a useful dataset contribution, providing a valuable set of observations, which can be very useful for high resolution snow transport modelling in cold regions. The manuscript is well-written and could be improved by cutting details about modelling and empirical method and reorganizing the text. I recommend publication in Earth System Science Data as a regular article in the data section after a moderate revision. Below are some specific comments and

[Figure]

suggestions for improving this manuscript:

1. The manuscript is too long. I would recommend authors to focus more on the data and less on modelling and empirical estimation of snow transport.

2. From the text it is not clear how data gaps were filled and quality controlled?

3. So many acronyms were used in the abstract and text that makes it difficult to follow the content. It is suggested to avoid unnecessary references (e.g., one in the abstract) and acronyms e.g., OZCAR, SAFRAN, CLIMate, OSUG, CNRS, ETNA, IRSTEA or Irstea (line 61), ARPEGE (166).

4. The main concern is that comparison of the two snow transport (frequency and mass) products, estimated by an empirical method and measured by Snow Particle Counter (SPC), is not one to one and I could not obtain the same conclusions that the authors have provided at the end of the section 2.3.3.

Editorial comments:

Abstract-Line 20: Remove "data" after "Observations".

Abstract: What does "SAFRAN" stand for? What is a local meteorological reanalysis?

Replace "altitude" with "elevation" in the manuscript. "Elevation" represents the position of the sites and stations better than "altitude".

Line 54: Which "last seasons"? Provide more details.

Line 65: replace "strong" with "large".

Figure 1: define all the subplots in the figure caption. Only subplot c is defined in the current caption. Add geographic names on subplot a.

I recommend Table 1 to be merged to Table 2.

Line 133-134: The "...from AWS Muzelle and Col are available from winter 2002-2003

and 2010-2011..." is not consistent with Table 2. Check this. Even it is recommended that you remove this as it is already repeated in Table 2.

Section 2.2.1 Wind speed and direction: I would avoid using indices or names for the sensors (e.g., lines 138, 141, 147, 157) in the text and mention them only in Table 2.

Line 168: replace "get" with "obtain".

---

## Author Comment (AC1) · 8 Oct 2018

**Response to Reviewer 1**

General comments:
This paper presents a meteorological database of blowing snow events for the Col du Lac Blanc study site a high-altitude experimental site located in Grandes Rousses range (French Alps). In-situ observations are obtained in four different automatic weather stations located within the study site. Additionally meteorological information is completed with SAFRAN model reanalysis. It is also described the methodology for obtaining blowing snow events and the data obtained with this methods are included in the database. The dataset described in this article has a great potential for many applications for studying snow dynamics on mountain areas. For this reason the manuscript should be published. Nevertheless there are some issues that must be addressed before its final publication.

We thank Reviewer 1 for his insightful comments. We answered below to all his points. His comments are in normal font while our answers appear in blue. Changes made to the original version of the paper appear in blue italic.

Major points:

1.- I encourage to include a new section explaining and describing Blowing snow data and the methods used (this is, change section 2.3 to section 3), since this probably is the most novel part of the paper. This section must clearly state from which AWS are the data. Sometimes it is difficult to follow this. Moreover Table 3 must be divided. You first present results obtained in section 2.3.2 with the particles thresholds. Afterwards you present a new "Table 4" with results shown in the two last rows of Table 3 since you are using there same method to compare the occurrence of drifting snow. This will help to understand the table faster for potential readers. I also miss some discussion about the fact that in Figure 4, when less data are available (percentage of valid data derived from SPC) more quantity of snow is detected and a higher percentage of time detected particles.

Following the recommendations of Reviewer 1, we added a new section 3 dedicated to blowing snow data. The content of this section has been modified following the comments of the three reviewers (see our answers to their different points). In particular, we improved the description of the 2 sources of blowing data: empirical database of blowing snow events and SPC measurements. Table 3 in the initial version of paper has been split into two tables. We hope that this revised presentation will help the potential readers. Concerning the last point, the percentage of valid data derived from SPC provides a confidence index in data (it is now indicated in Figure 4's legend). For example if we record data during only one day with snow storm throughout the season, we will obtain a low percentage of valid data (1%) with a high percentage of time during which particles are detected by SPC-S7 (100%) and a high quantity of snow (an average quantity of snow transported between 0.2 and 1.2 m per linear meter during 30 days (q*30) is calculated from the quantity (q) recorded during one day). As conclusion, there is no direct link between percentage of valid data delivered by SPC-S7 (which includes periods with and without blowing snow) with percentage of time when snow particles are detected by SPC-S7 (which only includes periods with blowing snow).

2.- I have missed some information about the climatic characteristics of the study site. As authors say, the experimental site has been operationally used since 1988. I think it is really interesting to provide an overview of the climatology observed in this site. For instance it could be included the mean annual and winter temperature, number of days with snow presence in the automatic weather station with the longest dataset, total annual precipitation:

The site has been operationally used since 1988 but was initially only dedicated to experimental campaigns dedicated to blowing snow processes and no continuous meteorological record of sufficient quality can be used to derive climatic characteristics of the site from 1988 to present. In the revised version of the paper, we included in Section 2.1 (Site characteristics) information on the winter climate of the site using the data available in the database to compute them. The revised paragraph is written as follows:

*"The Col du Lac Blanc (CLB) experimental site is located at 2720 m a.s.l. in the Grandes Rousses mountain range (45.13°N, 6.12°E, Fig. 1) in the central French Alps. The mean winter temperature (December-March from 2000 to 2016) at AWS Lac Blanc (Fig. 1) was -6.1 °C. The study site covers an altitudinal range between 2700 and 2800 m a.s.l and can be viewed as a natural wind tunnel due to its orientation and the specific configuration of the surrounding summits. Indeed, the Grandes Rousses range on the eastern side and the "Dôme des Petites Rousses" summit on the western side channel the atmospheric flow according to a North-South axis (Fig 1). This characteristic of the site is particularly useful for studies on the effects of wind on snow redistribution. Over the winter period (December-March) from 2000 to 2016, the mean wind speed at AWS Lac Blanc was 4.9 m s⁻¹ with a maximum wind of 38.3 m s⁻¹ reached on 28 January 2006. Wind speeds larger than 10 m s⁻¹ were recorded on average 10.2 % of the time in wintertime. Snow is typically present on the ground around the site (Fig. 2) from late October to early June. The mean winter snow depth (December-March) from 2000 to 2016 at AWS Lac Blanc was 1.88 m and presents a large inter-annual variability with a minimal value of 1.28 m in 2001/2002 and a maximal value of 2.83 m for winter 2003/2004. ...." ." ( 68 in the revised manuscript)*

3.- If possible, I encourage manuscript authors to include in the database observations obtained during the whole study period and not only during winter period. This can be really interesting since can provide an evaluation of observations/model deviations on an annual time basis. Moreover I think that observations of the last two snow seasons 2016-2017 and 2017-2018 are quite valuable, so I encourage manuscript authors to, upload this information during the review process.

The Col du Lac Blanc experimental site has been designed to primarily focus on processes influencing the wintertime evolution of the snow cover. For this reason, continuous meteorological records are not available for the period outside the winter season. In particular, in summertime, instruments undergo revision and calibration (if necessary). SAFRAN outputs are provided during the whole study period to overcome this limitation.
We totally agree with Reviewer 1 that the last two snow seasons 2016-2017 and 2017-2018 are quite valuable. They will be available with the same doi as specified in Section 4. However we had not enough time to upload this information during the review process. The fact that the data acquisition at the site still continues is now also mentioned at the end of the introduction.

4.- The is mostly focused on data obtained with different environmental sensors. This way along the manuscript the model and the company (including a reference to their data) of sensors must be specified.
Sensors models and company are now specified in Table 1 in order to avoid repetition throughout the text.

Specific comments:

Line 15: Precise that "Grandes Rouses" is located in French Alps. Correction included.

Line 20: Precise the period in which the Snow Particle Counter acquired observations (2010-2016). Correction included

Line 28: Remove Gaillardet et al., 2018 reference. It is not appropriate to include a reference of an article submitted, even more if it is included in the abstract.
This article is now accepted. The reference for this paper was been removed from the abstract and is now included in the introduction.

Line 43: Maybe rephrase as: ": : :have joined their efforts to investigate the effect of wind transport on snowpack evolution." Correction included.

Line 44: "A high-altitude experimental site WAS set up: : :" Correction included.

Line 45: By inspection of Figure 2. I guess that the study area covers an altitudinal range of about 200-300m. Please include maximum and minimum elevations in the text.
The study area covers an altitudinal range of approximately 100 m between 2700 and 2800 m (Fig. 1c). It is now mentioned in the text.

Line 53-55: Change appropriately in regard to the major comment of including a new section for describing blowing snow data.
Correction included.

Line 60: Describe the locations of Grandes Rouses within the Alps and include the altitudinal range of the study site. We now mentioned in the text that the Grandes Rousses range is located in the central French Alps. Its exact location is shown on Fig. 1. The altitudinal range between 2700 and 2800 m a.s.l is also given.

Line 73: In table 1 and line 61, you provide the location of the automatic weather station on longitude, latitude; could you please also provide these coordinates on same coordinate system of the DEM available in the database?
The exact AWS locations are now also given in Table 1 in Lambert 93 coordinates, the same coordinate system as the DEM. We did not include them in the text when giving the general location of the Grandes Rousses range in lat/lon.

Line 130: Which is the "manual quality check" process? You remove outliers?
The procedure of quality control is now described for each variable in the revised version of the manuscript. Outliers were removed from the final datasets and replaced by nan values in the csv files.

Line138: Which young sensor? There are several products of this company. A Young 05103 wind monitor is installed at the AWS Lac Blanc. It is now explicitly included in Table 1.

Line 142: There is a final "." missed. Correction included.

Line 147: Include the company of PT100 wires. Correction included in Table 1

Line 149: You have already said that height changes during the course of the winter. Additionally you don't provide the snow free-height of these sensors. Remove this sentence.
As suggested by Reviewer 1, we removed the first sentence mentioning the changing measurement height during the winter. However, we decided to keep the second sentence since the snow-free height of each sensor is given in Table 1 (6th column).

Line 152: Please, include model and company of the ultra-sound snow height sensor. I also suggest giving a small explanation about how these sensors work.
Model and company of the ultra-sound snow height sensors are now indicated in Table 1. The following sentences were added to describe how the sensors work:
*"These sensors determine the distance to the snow surface by sending out ultrasonic pulses and listening for the returning echoes that are reflected from the surface. The time from transmissions to return of an echo and the speed of sound in the air are then used for obtaining the distance measurement. Since the speed of sound in the air varies with air temperature, a correction of the distance calculation is carried out using the air temperature measurements previously described." (L 152 in the revised manuscript)*

Line 155: The surface area of the ultra-sound sensor observation may variate depending on snow height. Please clarify and quantify maximum and minimum surface area values.
We added the following sentence to quantitatively describes how the surface area observed by the sensor changes as a function of snow depth:
*"The beam angle is 30° which gives an observed surface area of 8 m² for a sensor mounted at 6 m above a snow-free ground (the average sensor height at AWS Muzelle and Lac Blanc, Table 1). This*

*area decreases down to 0.9 $m^2$ when the snow depth reaches 4 m." (L 157 in the revised manuscript)*

Line 157: Include the company of SHM30 sensor. The name of the company is now indicated in Table 1.

Line 164: In the abstract you said that you provided SAFRAN reanalysis and here it is said that you provide SAFRAN analysis. Please clarify.
We thank for Reviewer 1 for identifying this inconsistency. Since we provided outputs form the SAFRAN reanalysis, we modified the text accordingly.

Lines 164-176: I see quite interesting to include SAFRAN model outputs. If I am right, this is not a 2D model. Please explain how you obtain the data for Col du Lac Blanc.
Indeed, SAFRAN is not a 2D model. It is a meteorological application that uses a conceptual representation of the topography by elevation bands for specific mountainous area known as "massifs". The following sentences were added:
*"They are available per 300-m elevation bands for areas known as massifs (23 in the French Alps) which were defined for their climatological homogeneity (Durand et al., 1993). To obtain SAFRAN data at the elevation of CLB (2720 m a.s.l), we used a weighted-average of the data from the Grandes Rousses massif for the elevation bands 2700 m and 3000 m." (L 170 in the revised manuscript)*

Line 173: You say SAFRAN is considered as the reference precipitation in Col du Lac Blanc; however this is a model and could have errors. Please discuss this issue and provide an estimation of potential bias of this model (even if it is for a different study area) in the Alps.
SAFRAN is a meteorological analysis system which combines for each massif a climatological precipitation gradient with all the measurements of daily precipitation available in the massif using optimal interpolation. For the Grandes Rousses, in mid-winter, 6 stations are used for the analysis covering an altitudinal range between 1350 and 2350 m. The main sources of uncertainty in the precipitation analysis are: (i) the quality of the precipitation measurement potentially affected by wind under catch, (ii) the shape of the climatological precipitation gradient, especially at high-elevation where no station are used in the analysis. The precipitation analysis at the elevation of Col du Lac Blanc is affected by these two sources of uncertainties. Previous studies compared SAFRAN seasonal snowfall and measurements of winter mass balance for the St Sorlin glacier located in the Grandes Rousses range between 2650 and 3400 m (Gerbaux et al., 2005; Dumont et al., 2012; Reveillet et al., 2018). They reported an underestimation of SAFRAN seasonal snowfall ranging between 33 and 42 %. In the revised version of the paper, the section on the SAFRAN analysis has been rewritten as follows:

*"SAFRAN data at CLB are provided for all the meteorological variables required to run continuously a land surface model at CLB without data gap for the entire time period.. SAFRAN data includes 2-m wind speed, 2-m air temperature and humidity, incoming longwave and shortwave radiation and snowfall and rainfall amount at an hourly time step. Using SAFRAN data for driving a land surface model at CLB requires accounting for several limitations of this dataset, in the interpretation of the results, especially for solid precipitation and wind speed and direction. Directly measuring solid precipitation at CLB has been abandoned for several years, because of the strong undercatch under windy conditions (e.g. Kochendorfer et al., 2017) which could not be addressed adequately in-situ. Instead, SAFRAN data are used for solid precipitation, which uses observations at neighboring sites.*
*Note however that the accuracy of the SAFRAN precipitation data is known to be limited at high elevation such as CLB, because of the restricted number of stations incorporated into the analysis above 2000 m (only 2 in the Grandes Rousses range) and the potential influence of wind undercatch at these stations. Gerbaux et al. (2005), Dumont et al. (2012) and Reveillet et al. (2018) compared SAFRAN total solid precipitation amounts at the annual scale and observed winter mass balance of the St Sorlin glacier (2650 m -3400 m) located in the Grandes Rousses 5 kilometres away from CLB. They found an underestimation of SAFRAN winter precipitation ranging between 33 and 42 %. At CLB, the underestimation is expected to be less since glacier are generally preferential accumulation areas for a given elevation.*

*Wind speed is generally underestimated by SAFRAN CLB due to the influence of the surrounding topography which is not included in the conceptual representation of the topography in SAFRAN. It is recommended to replace SAFRAN wind speed and direction by the observations collected at CLB when running a land surface scheme at CLB, as described in Vionnet et al. (2013)."*

Line 195-204: These sentences are difficult to follow, please rephrase. For instance when you say: "Positive values of the difference: : :." I think you are describing the method you refer before as "This indirect method: : :" but this is not clear.

This part of the paper has been rewritten to be easier to follow:

*« Periods of ground snow transport were then identified at an hourly time step from an analysis of the recordings from the snow depth sensor. Positive values of the difference between the maximum and minimum snow depth recorded over an hour and associated large values of its standard deviation are characteristic of the presence of snow particles between the sensor and the surface of the snowpack. Snow particles in motion above the snowpack surface create indeed interference in the ultrasonic signal. This indirect method to identify blowing snow occurrence was developed and tested over fifteen years of observations at Col du Lac Blanc. » (L 212 in the revised manuscript)*

Line 203: How did you complete the analysis for the period 2000-2004 without the webcam? Maybe you could explain that the results obtained for the period 2004-2016 were evaluated with a visual inspection of webcam images.

The sentence has been modified based on the recommendation of Reviewer 1:

*"The results obtained for the period 2004-2016 were evaluated with a visual inspection of webcam images, in particular to improve the identification of blowing snow events with and without concurrent snowfall." ." (L 218 in the revised manuscript)*

Line 204: Change "recorded" by "included". Correction included.

Line 225: Why 917 kg m$^{-3}$ density value?

The snow particles blow as rounded grains, not snowflakes. Indeed, the saltation process will quickly round the edges of original snow crystals and the grains become well rounded and are assumed to be spherical. The accuracy of mass flux measurement depends on blowing snow characteristics. Thus the particle density is set to the ice density and is equal to 917 kg m$^{-3}$. Blowing snow accompanied by snowfall contains not only spherical particles but also variously shaped particles including snow crystals and snowflakes, and its size distribution extends to a larger range.

Explanation is now included in the revised manuscript *(L 245 in the revised manuscript)*

*"They blow as rounded grains, not snowflakes. Indeed, the saltation process quickly rounds the edges of original snowflakes and the grains become well rounded and are assumed to be spherical. The accuracy of mass flux measurement depends on blowing snow characteristics. Thus the particle density is set to the ice density and is equal to 917 kg m$^{-3}$."*

Line 228: Here you use the acronym SPC and not anymore SPC-S7s as you did before. Please be consistent along the manuscript when you refer to this device.

The term "SPC-S7" is now used throughout the text.

Line 240: Include A, z and m values you used to estimate mean horizontal flux and its vertical interpolation.

All available data (i.e. at different heights z) are used to estimate the mean horizontal flux at 1-m and its vertically-integrated value. A and m are not constant over time and are calculated at each 10-min time step. It is now specified *(L 265 in the revised manuscript)*

Line 220 and 238: include a reference for mathematic equations (1, 2: : :). Numbering included.

Line 251: I guess this is the power law you introduce in line 238. Use a number to refer this expression. Correction included.

Line 263 to 264: When you present "kg" of snow, specify that you are showing snow mass transport variable.

We are not sure to clearly understand the comment and we replace "quantity of snow transported" by "blowing snow transport quantities…."

Line 266: ": : :to keep in mind that SPC, which detects each particle, is able to: : :" Correction included.

Line 267: This is discussed in next section in several paragraphs: Correction included.

Line 278: You already showed the 50% if time and the 6245 kg of mass on previous section. This is redundant. You can remove it here. It has been removed.

Line 291: I guess these conclusions came from table 3 since Figure 4 does not show the empirical method. Remove Figure 4 reference.
Correction included

Line 299: I find quite surprising that the occurrence of wind-induced snow transport is closer to 30% of the time. Has been shown this value before? Where?
It is not so surprising if we consider that snowfall events with light wind are also recorded by the SPC. Higher transport frequency with seasonal variation have been observed in Adélie Land (Trouvilliez et al., 2015 http://dx.doi.org/10.1016/j.coldregions.2014.09.005)

Line 301: You mean the empirical method with SPC data? Please clarify.
Empirical method systematically refers to Empirical database of blowing snow occurrence. In the new version, we used the same name throughout the text to avoid confusion.

Line 316: Include mean snow depth value during the 2010-2016 time period.
It is now included in text:
*"The mean snow depth value during the 2010-2016 period was 1.99 m."* (L 349 in the revised manuscript)

Section 4, data availability: The database must include a metadata file for each AWS that includes all variables of each file, their units and the location of the station. Moreover I think it is not necessary to include the doi of each single station, for SAFRAN reanalysis and for the DEM all these links can be easily found following the link: http://doi.osug.fr/public/CRYOBSCLIM_CLB/CRYOBSCLIM.CLB.all.html
It was already done in the original version but not easy to find if the reader use only the link http://doi.osug.fr/public/CRYOBSCLIM_CLB/CRYOBSCLIM.CLB.all.html as suggested by the reviewer. That's why we prefer to include the doi for each single station. But we fully understand reviewer's viewpoint and for greater clarity, dois are now presented in a new table (Table 4).
The metadata are clearly described when accessing the doi for each dataset and includes the variables of each files, their units and the location of the station.

Concerning the DEM, it must be provided DEM on a single file and not in 14 separate files. The research group knows in detail the study site characteristics and any incoherence coming from alignment errors of the separate files can easily be detected, what it is not the case for potential users of the DEM. If necessary the spatial resolution of the DEM can be reduced to 0.5 m or 1 m grid cell size. X, Y and Z units and column names must be included in xyz files. Also a metadata diles of the DEM must be included.
The DEM is now provided as a single file with a spatial resolution reduced to 1 m to be easy for the user. The metadata for the DEM are described when accessing the corresponding doi.

Figures and tables:

Figure 1: Please include in c) panel the "Dôme des Petites Rouses" triangle and the point that marks "Col du lac Blanc" from b) map. I also encourage manuscript authors to draw dashed lines on c) map showing the area covered in picture of Figure 2. This would really help to potential readers to understand the characteristics of the study site.

Figure 1 has been modified following the recommendations of Reviewer 2. We also include the following suggestions made by Reviewer 1. The "Dôme des Petites Rouses" triangle is now shown on map c) whereas the point that marks "Col du lac Blanc" has been removed from map b). Instead, the location of the four AWS is shown on map b). A blue dashed line depicts the approximate contour of the area covered by the picture shown in Fig. 2. The legends of Fig. 1 and 2 have been modified accordingly.

Figure 5: I guess that the different circles of wind roses show the frequency of the different events. Please clarify.

The different circles shows the frequency and the values of frequency are given on the graph.

Maybe it is more interesting to provide the wind rose for AWS Col (same of (b) wind rose) since you are showing the fluxes obtained in this station.

The wind rose for AWS Col is now provided and is consistent with the rose showing blowing snow fluxes.

Table 2: Please group first column when same station is described. It will be easier to understand the table.

In the revised version of the manuscript, we merged Table 1 and 2 as recommend by Reviewer 2 and group the first column when the same station is described.

Table 3: For the second column of the table, put "Threshold" in the first cell and not for each single cell. Units of the different variables are not appropriately included; in some cases the occurrence of drifting snow presents the%in others not. Similarly Kg of snow mass transport are not provided. Please be consistent along the table. Moreover there is a mistake and the 2p/cm2/min threshold is a 20p/cm2/min threshold as introduced in line 248. In the last row of table 3: You say that it is shown the "Total quantity of snow transported" however I think it is the occurrence of drifting snow. Moreover I see a bit confusing that you show in this row the results obtained with the method presented in section 2.3.1 with the SPC data without introducing this before.

The new section 3 has been rewritten and all the comments suggested by Reviewer 1 taken into account.

Supplementary material: Line 20: Remove one "of" right before 500*500.

Sentence has been removed.

---

## Author Comment (AC2) · 8 Oct 2018

**Response to Reviewer 2**

The paper describes the meteorological and blowing snow dataset collected at Col du Lac Blanc (2720 m a.s.l.) in the French Alps. This data consists of wind speed and direction, air temperature, snow depth, blowing snow fluxes and occurrence periods, spanning the period from 2000/2001 until 2015/2016. The data is complemented with local atmospheric reanalysis from SAFRAN, and a digital terrain model with 20 cm resolution. All data are now available for the public, the dog's are given. The paper is a very welcome contribution for the ESSD Special Issue: Hydrometeorological data from mountain and alpine research catchments, and for the scientific community of snow scientists. However, I recommend some improvements prior to final publication.

We thank Reviewer 2 for his comments that greatly helped us improving the quality of the paper. We answered below to all his points. His comments are in normal font while our answers appear in blue. Changes made to the original version of the paper appear in blue italic.

General aspects:

- my major concern is that in the way it is presented, the comparison of the database of blowing snow occurrence and the SPC is not meaningful (2.3.1, 2.3.2 and 2.3.3). Both data sources make use of more or less empirical thresholds to classify a period as blowing snow occurrence, or to count the percentage of time during which particles are detected by the SPC, respectively. There is no better or worse of the two methods, they are only different and, due to the threshold values chosen, provide different results. Why do you choose thresholds in a way that the results become such different? The possibilities for improvement that I see here are (i) just present the two datasets "as is" without comparison, (ii) explicitly justify the choice of all thresholds, and explain the difference in the results, (iii) calibrate one method with the other, or (iv) leave the empirical database out und just provide a reference. In any case, always distinguish clearly physical processes from empirical estimates. Finally, it is not clear who the original author of the database methodology is, Guyomarc'h and Merindol (1998) or Vionnet et al. (2013)?

Following the recommendations of Reviewer 1, we have added a new section 3 dedicated to blowing snow data with new comments and we have split Table 3 in two tables. We hope that this revised presentation will help the potential readers:

- Section 3.1 describes the empirical database of blowing snow events. The choice of the minimum duration of 4 hours is better explained (see below our answer to this specific point)

- Section 3,2 describes the SPC measurements. The last paragraph of this section (L 231) has been fully rewritten to better explain the impact of the threshold value applied on the particle flux. Such discussion is important since SPC data are not widely used in the community

*"Figure 4 shows an overview of the inter-annual variability of blowing snow occurrence and intensity derived from the SPC-S7. They measured non-zero snow flux during a percentage of time ranging from 43 % in 2010-2011 to 63 % in 2013/2014 corresponding to an average blowing snow transport quantities between 0.2 and 1.2 m per linear meter during 30 days ranging from 2547 kg in 2010-2011 to 10331 kg in 2013/2014. Overall, for the period 2010-2016, SPC-S7 provided a non-zero-value during 50% of time and an average quantity of snow transported between 0.2 and 1.2 m per linear meter and per 30 days of 6245 kg. However, SPC-S7 can detect individual snow particles and can report a positive signal (particle number larger than 20 particles cm⁻² during 10 min) even during very light snowfall event with low wind. Such period is considered as a blowing snow period even if it does not significantly contribute to the total amount of transported snow. For this reason, we quantified the sensitivity of the estimations of blowing snow occurrence and amount to the threshold value used to set flux to zero. Results are presented on Table 2. Three threshold values were tested: (i) the initial value of 20 particles per cm² per 10 minutes used to remove to electronic noise, (ii) a threshold value 1200 particles cm⁻² per 10 minutes as in Naaim-Bouvet et al. (2014 ) and (ii) a value of 9600 particles cm⁻² per 10 minutes. Table 2 shows that the estimation of blowing snow occurrence*

*is highly dependent on the threshold values whereas blowing snow quantities remain quite stable in the chosen range. It is therefore essential to provide the chosen threshold value to end-users when determining blowing snow occurrence from SPC-S7. In general, it might be better to use blowing snow fluxes for characterizing blowing snow events".*

- Sections 3.3 (L 312) describes the comparison between the empirical database of blowing snow occurrence and the SPC. This section has been rewritten to provide a more accurate analysis of the results presented in Table 3. We decided to maintain this comparison because, from our point of view, it is useful for end-users. They need to be aware of the advantages and disadvantages of each method to use the data in the most efficient way. Moreover the depth of data available is greater for empirical data base at Col du Lac Blanc: 11 additional years are provided (2000-2010). We therefore add a new paragraph dealing with the potential use of data.

*"The estimation of blowing snow occurrence determined with the SPC-S7 reported on Table 2 differs from the results obtained with the empirical database (Fig. 3). To gain more understanding on these differences, we determined the quantity of snow transported between 0.2 and 1.2 m per linear meter during the periods identified as blowing snow periods in the empirical database and we compared this value with the total quantity of snow transported between 0.2 and 1.2 m per linear meter derived with the SPC-S7 for the same winter season. The result is expressed as a percentage in Table 3. It shows that the empirical database of blowing snow occurrence detects 55 % of the total transported snow mass measured by the SPC-S7. This results from the non-detection of blowing snow events of low to moderate intensity with the empirical method as discussed in Vionnet et al. (2013). This method only reports the main blowing snow events. This mainly results from assumptions made in the method: the minimal event duration is set to 4 hours and only period with wind speed greater than 6 m s-1 are included during snowfall. Therefore, the estimation of blowing snow occurrence with the empirical method (12.0 % of the time for the period 2010-2016; Table3) constitutes a lower bound for the estimation of blowing snow occurrence at CLB. SPC-S7 provides estimations ranging between 23 and 50 % of the time, depending on the threshold value used when filtering the SPC-S7 data as discussed in the previous section.*

*The empirical database of blowing snow events and the SPC-S7 data are two sources of information on blowing snow occurrence and intensity at CLB. We recommend the use of SPC-S7 data for the study of blowing snow processes and the evaluation of models at fine temporal scales whereas the empirical database of blowing snow events can be used to evaluate reanalysis or output of regional climate models on a longer term. Compare to the SPC-S7 data, the empirical database covers a longer time period (11 additional years: 2000-2010). It also provides continuous hourly estimations of blowing snow occurrence whereas about 25 % of the SPC-S7 data can be considered as invalid or missing over the period 2010-2016 (Fig. 4)".*

The empirical method to determine blowing snow occurrence has been developed by Guyomarc'h and Merindol (1998) and used in Vionnet et al. (2013). It is now clearly stated in the revised version of the paper (L215)

- my second major concern is a thorough discussion of the (very important) scale effects of blowing snow, and which ones are observed at Col du Lac Blanc. This should be an original paragraph in the introduction

The introduction has been rewritten to mention the different scale effects of blowing snow. Rather than writing an original paragraph, we included it at 2 different stage of the introduction.

- In the first paragraph, when presenting in general the effects of blowing snow on snow cover variability:

*"Snow deposition is affected by wind-induced snow transport at the slope scale in a wide scale range of few meters to hundreds of meters (e.g. Mott et al., 2010). Snow tend to be deposited in the lee of ridges or local depressions leading to the formation of snow dunes and drifts and the smoothing of the land-surface roughness (Schirmer and Lehning, 2011). Blowing snow also affects the surface roughness of the snow cover at sub-meter scale creating a large variety of Aeolian snow forms such as ripples or sastrugis (e.g. Filhol et al., 2015)."(L32)*

- In the second paragraph, when describing the effects observed at CLB:
*"Wind-induced snow transport strongly modifies the spatial variability of the snow cover around the site for scales ranging approximately from 1 m to 50 m (Vionnet et al., 2014; Schön et al, 2015, 2018). The roughness of the snow surface is also continuously evolving throughout the winter as a function of the occurrence of snowfall and blowing snow events (Naaim Bouvet et al. 2017)." (L47)*

- the fact that the data collection continues and new measurements will follow and be made available should be mentioned in the beginning of the paper, not at the end (in section 4).
It is now also mentioned in the introduction section. (L61)

- in figure 1 (the maps) the color scheme/contour lines should be improved: certain altitudes should be associated with a contour line, not with a color. The latter should be associated with an altitudinal range. In this map figure, the typeface and size should be harmonized. In the legend, "Automatic stations" should be "Automatic weather stations", or simply "AWS"
New versions of Maps (b) and (c) on Figure 1 have been produced during the review process. They include an updated color scheme, updated contour lines as well as harmonized font size and type. The term "AWS" is now used on Figure 1. The figure caption has been modified accordingly.

- Table 3 is more confusing than helpful; is there not better way of presenting these numbers?
Following the recommendation of Reviewer 1, we have split Table 3 into two tables and we delete lines which don't provide meaningful information. In such way, we believe the two new tables will provide valuable information to the readers.

- the reference section is full of type inconsistencies and mistakes. This entire section needs complete revision (should have been checked prior to submission).
The reference section has undergone a careful check to remove all inconsistencies and mistakes.

- the use of lowercase and uppercase letters needs revision in the entire manuscript

Details:

- line 20: "Snow Particle Counter" in uppercase: why? Is this the name of the device? I would recommend "snow particle counter (SPC)", and use "SPC" only in the following text.
Correction included. SPC-S7 is used in the rest of text.

- line 21/22: give the date when resolution changes Date included.

- line 28: avoid references in the abstract The reference was suppressed in the abstract.

- line 33: insert "atmospheric" between "concurrent" and "snowfall" Correction included.

- line 51: "SPC" SPC-S7 which is a commercial name is used in the text after the description of the sensor (Table 1)

- line 72: insert "weather" between "automatic" and "stations" Correction included.

- line 90: better "a" instead of "the" ("detailed view") Correction included.

- Table 1: remove the dot after "direction" Correction included.

- Table 2: remove the dots in the units, insert space in "4.8m" Correction included.

- line 133: the availability of the AWS Muzelle data given here does not match the one given in table 2
We thank Reviewer 2 for pointing out this inconsistency. Temperature and snow depth data are available at AWS Muzelle since December 2004 whereas wind data are available since December

2002. The text and the Table 2 have been modified to correct the inconsistency.

- line 1390: remove dot in "m.s-1" Correction included.

- line 142: insert dot after "process" Correction included.

- line 145: replace "Sect." We kept the term "Sect" following the recommendation for authors available on the ESSD website.

- line 162: better remove "but its power consumption is higher" - this is another issue of no relevance here Sentence removed as recommended.

- entire section 2.2.4: make clear if the SAFRAN data presented here origins in an "analysis", or in a "re-analysis", and use the correct term then everywhere
The term reanalysis is now used everywhere.

- line 181: "SPC" Correction included.

- line 183. insert "snow" between "blowing" and "occurrence" Correction included.

- line 184: replace "of the period" with "in the period", and add at the end of the sentence where the sensor was established. Replace "consists in" with "consists of"
The text was modified following this comment. The location of the sensor was not added since the database of blowing snow events is derived from a combination of sensors as explained at L 191-205 in the initial version of the manuscript.

- line 186: replace "was" with "is", make "event" plural ("events") Correction included.

- line 188: better "relies" instead of "relied", and "requires" instead of "required" Correction included.

- line 189: replace "datasets" with "meteorological occurrences"
We think that the term "datasets" is more appropriate in this case. Therefore we did not include the modification.

- line 191: better formulate "Periods of ground snow transport with concurrent snowfall are identified first" Correction included.

- line 196: do you mean a logarithmic law? If yes, write the entire word and avoid abbreviations
We replaced "log-law" by "logarithmic law".

- line 204: how did you choose the threshold of 4 hours? The results depend on it!
The empirical database of blowing snow occurrence has been developed to provide a full record of blowing snow occurrence that can be compared with the avalanche activity in the Grandes Rousses massif (Guyomarc'h et al., 2014). For this reason, the threshold of 4 hours has been selected to only keep in the database the main blowing events. We are totally aware that this is a strong limitation when compared to the occurrence of blowing snow measured by the SPC.
In the revised version of the manuscript, we better justify the choice of this 4-hour threshold:

- *"This database was initially developed to provide a full record of blowing snow occurrence that can be compared with the avalanche activity in the Grandes Rousses massif (Guyomarc'h et al., 2014)." L199 in the revised manuscript)*
- *« Only events of duration longer than 4 h were recorded in the database to only include the main blowing events." (L 221 in the revised manuscript)"This method only reports the main blowing snow events. This mainly results from assumptions made in the method: the minimal event duration is set to 4 hours and only period with wind speed greater than 6 m s$^{-1}$ are included during snowfall." (L 322 in the revised manuscript)*

- line 207: does this correspond to the sum of the two columns in a winter season? Indicate this in the text. Avoid to write "blowing snow occurred" in the context of the empirical method with the threshold, since the process of blowing snow probably occurred much more often! (i.e., write something like "blowing snow periods were classified : : :")

*Yes, the values correspond to the sum of the 2 values reported on Fig 3. It is now described in the revised version of the manuscript:*

*"These values correspond to the sum of blowing occurrence with and without concurrent snowfall reported on Fig. 3." (L226 in the revised manuscript)*

*The sentence "blowing snow occurred" has been removed from the text and we now use:*

*"Using this method, blowing snow periods were identified during 11.7 % of time at Col du Lac Blanc over the period 2000-2016. 36.7 % of time blowing snow periods were classified with concurrent snowfall." (L 227 in the revised manuscript)*

- line 209: why only "are similar", should these estimations not be the same as in Vionnet et al. (2013)? What is different, and why?

*We used "are similar" since the estimations in Vionnet et al. (2013) covered the period 2000-2011 whereas the database presented in this paper contains 5 more years and covers the period 2000-2016. We used in the revised version of the paper:*

*"These estimations for the period 2001-2016 are similar to those reported in Vionnet et al. (2013) for the period 2001-2011." (L 228 in the revised manuscript)*

- line 214: better write "Data from SPC devices" *Correction included.*

- line 215: "SPC" *Correction included.*

- line 216: insert "a" between "on" and "photodiode" *Correction included.*

- line 216/217: better make two sentences out of the one *Sentence modified*

- line 226: "SPC" *Correction included.*

- line 227: delete "and risked burying the sensors" *Sentence deleted.*

- line 228: better of "the" SPC *Correction included.*

- line 229: delete "being made". Better write full words, i.e. "That is why" *Correction included.*

- line 235: insert "the" between "approximate" and "averaged" *Correction included.*

- line 338: correctly align the minus in the formula *Correction included.*

- line 244: delete "case of", add "speed" (?) after "wind" *Correction included.*

- line 248: delete the "s" at the end of "particles" (-> singular) *Correction included.*

- line 258: better "the mean horizontal fluxes" *Correction included.*

- line 261: "the" SPC data *Correction included.*

- line 263: "The" SPC *Correction included.*

- line 265: "in" the period *Correction included.*

- line 266: "the" SPC *Correction included.*

- line 266: : : : able to "identify trace" precipitation : : : ? Please clarify!
The term "trace precipitation" is generally used to describe a very small amount of precipitation that results in no measurable accumulation using a rain gauge, snow stick, or any other weather instrument. (http://glossary.ametsoc.org/wiki/Trace). As SPC-S7 is able to detect one snowflake, it is able to detect trace precipitation. The expression "trace precipitation" has been deleted in the text to facilitate reader comprehension *(L 299 in the revised manuscript)*
*"However, SPC-S7 can detect individual snow particles and can report a positive signal (particle number larger than 20 particles cm-² during 10 min) even during very light snowfall event with low wind".*

- Figure 4: add unit to y-axis and format typesetting properly.
We added the unit to the y-axis.

In the figure caption, better write "during" the period, and "delivered by the SPC": Corrections included

-  line 279: this "calls" for Correction included.

- line 281: better "That is why" Correction included.

- line 282: "applied for the raw data", with "a particle flux smaller : : :" (singular) Correction included.

- line 283: "Then a similar data processing: : :", and: why "similar", and not "the same". Explain the difference, if it exists! It is not similar, it is the same. Correction is included.

- line 284: "a" direction threshold. Delete "parameters" Correction included.

- line 285: "with different thresholds" Correction included.

- line 293: "to the SPC" Correction included.

- line 295: "blowing snow occurrence" Correction included.

- line 309: site "for the last six snow seasons" Correction included.

- line 334: better "of" the different stations Correction included.

- line 341: "The data of the winters: : :" Correction included.

- line 350: "SPC" Correction included

- line 358: delete comma Correction included.

- line 359: better "who" instead of "which" Correction included.

Good luck, a very nice peace of work!  Thanks!

---

## Author Comment (AC3) · 8 Oct 2018

**Response to Reviewer 3**

General comment:
This manuscript presents wind, snow depth, and air temperature measurements from four automatic weather stations and measurements of blowing snow transport in a high elevation site in the Col du Lac Blanc, France. This is a useful dataset contribution, providing a valuable set of observations, which can be very useful for high resolution snow transport modelling in cold regions. The manuscript is well-written and could be improved by cutting details about modelling and empirical method and reorganizing the text. I recommend publication in Earth System Science Data as a regular article in the data section after a moderate revision. Below are some specific comments and suggestions for improving this manuscript:

We thank Reviewer 3 for his insightful comments. We answered below to all his points. His comments are in normal font while our answers appear in blue. Changes made to the original version of the paper appear in blue italic.

1. The manuscript is too long. I would recommend authors to focus more on the data and less on modelling and empirical estimation of snow transport.
There are differences of points of view between the reviewers about this question (see for example the first comment of reviewer 1: "I encourage to include a new section explaining and describing Blowing snow data and the methods used (this is, change section 2.3 to section 3), since this probably is the most novel part of the paper."). During the review process, we chose to follow Reviewer 1's advice and clearly described the blowing snow data since they are innovative. No numerical modelling is presented in this paper. As explained in the text, we simply applied regression on the SPC-S7 data to provide a consistent dataset throughout wintertime and to avoid the limitations associated with the changing height of the SPC-S7 sensors above the snow surface. The empirical database has clear limitations that are discussed in the text but we believe it provides valuable information to evaluate reanalysis or output of regional climate models.

2. From the text it is not clear how data gaps were filled and quality controlled?
No gap filling was performed during the process of quality control. Measurements identified as erroneous were simply removed from the dataset and replaced by nan values in the csv files. Sentences describing the quality control for each variable were added in the revised version of the manuscript:

- Wind: *"The temporal consistency of the wind speed data from the different AWS was controlled. Wind data (speed and direction) were removed from the datasets during periods of icing of the sensors identified with a visual inspection of webcam images for the period 2004-2016 and in the reports from the operators visiting CLB on weekly basis and thanks to a comparison between heated and non-heated anemometers data."* L139 in the revised version of paper.
- Temperature: *"Times series of air temperature were visually inspected and outliers removed from the final dataset, in particular when suspicious heating of the temperature sensors was identified during springtime periods with low wind and high solar radiation."* L147
- Snow depth: *"Time series of snow depth measurements were visually inspected and outliers removed from the dataset, most often occurring during snowfall."* L155 in the revised version of paper.
- Blowing snow fluxes: *The size distribution of blowing snow particles at a given height is represented by a gamma density function (Nishimura and Nemoto, 2005). That's why the size distribution recorded by SPC-7 is used to assess the temporal consistency of the blowing snow fluxes. Moreover calibration is performed prior to use at the beginning of winter season.* L249 in the revised version of paper.

3. So many acronyms were used in the abstract and text that makes it difficult to follow the content. It is suggested to avoid unnecessary references (e.g., one in the abstract) and acronyms e.g., OZCAR, SAFRAN, CLIMate, OSUG, CNRS, ETNA, IRSTEA or Irstea (line 61), ARPEGE (166).

The reference to Gaillardet et al. (2018) has been removed from the abstract. The acronym SAFRAN has been also removed from the abstract and its meaning is explained at the beginning of Sect. 2.2.4. CRYOBSCLIM, OSUG, OZCAR have been removed from the abstract.

Finally, we've tried to do our best concerning acronyms (ARPEGE was removed for example) but we can not ignore institutes or organizations which finance our salaries (CNRS, ETNA, IRSTEA – affiliation), support our research (OZCAR) or host the data (OSUG) even if they are not internationally recognized.

4. The main concern is that comparison of the two snow transport (frequency and mass) products, estimated by an empirical method and measured by Snow Particle Counter (SPC), is not one to one and I could not obtain the same conclusions that the authors have provided at the end of the section 2.3.3.

The revised version of the paper includes a new section only dedicated to blowing snow data (Section 3). In particular, the sub-section describing the comparison between the 2 sources of blowing data (empirical database of blowing snow events and SPC measurements) has been fully rewritten to provide a more accurate analysis of the results presented in Table 3. It is written as follows:

*"The estimation of blowing snow occurrence determined with the SPC-S7 reported on Table 2 differs from the results obtained with the empirical database (Fig. 3). To gain more understanding on these differences, we determined the quantity of snow transported between 0.2 and 1.2 m per linear meter during the periods identified as blowing snow periods in the empirical database and we compared this value with the total quantity of snow transported between 0.2 and 1.2 m per linear meter derived with the SPC-S7 for the same winter season. The result is expressed as a percentage in Table 3. It shows that the empirical database of blowing snow occurrence detects 55 % of the total transported snow mass measured by the SPC-S7. This results from the non-detection of blowing snow events of low to moderate intensity with the empirical method as discussed in Vionnet et al. (2013). This method only reports the main blowing snow events. This mainly results from assumptions made in the method: the minimal event duration is set to 4 hours and only period with wind speed greater than 6 m s$^{-1}$ are included during snowfall. Therefore, the estimation of blowing snow occurrence with the empirical method (12.0 % of the time for the period 2010-2016; Table3) constitutes a lower bound for the estimation of blowing snow occurrence at CLB. SPC-S7 provides estimations ranging between 23 and 50 % of the time, depending on the threshold value used when filtering the SPC-S7 data as discussed in the previous section.*

*The empirical database of blowing snow events and the SPC-S7 data are two sources of information on blowing snow occurrence and intensity at CLB. We recommend the use of SPC-S7 data for the study of blowing snow processes and the evaluation of models at fine temporal scales whereas the empirical database of blowing snow events can be used to evaluate reanalysis or output of regional climate models on a longer term. Compare to the SPC-S7 data, the empirical database covers a longer time period (11 additional years: 2000-2010). It also provides continuous hourly estimations of blowing snow occurrence whereas about 25 % of the SPC-S7 data can be considered as invalid or missing over the period 2010-2016 (Fig. 4)."* (L 316) in the revised version of paper.

Editorial comments:

Abstract-Line 20: Remove "data" after "Observations". Correction included

Abstract: What does "SAFRAN" stand for? What is a local meteorological reanalysis?

SAFRAN stands for Systeme d'Analyse Fournissant des Renseignements Atmospheriques a la Neige ; Analysis System Providing Atmospheric Information to Snow. We removed the acronym SAFRAN from the abstract and add the meaning of the acronym in Section 2.2.4.

We initially used the term « local » since SAFRAN outputs are only available in the French mountains but we agree that this term in not appropriate without a proper explanation. For this reason, we removed it from the abstract.

Replace "altitude" with "elevation" in the manuscript. "Elevation" represents the position of the sites and stations better than "altitude".

We replaced "altitude" by "elevation" in the revised version of paper. In particular, we changed the title of the paper.

Line 54: Which "last seasons"? Provide more details.

We replaced "over the last 6 seasons" by "from winter 2010-2011 to 2015-2016".

Line 65: replace "strong" with "large". Correction included

Figure 1: define all the subplots in the figure caption. Only subplot c is defined in the current caption. Add geographic names on subplot a.

The new caption of Figure 1 defines all the subplots and includes modifications based on comments made by Reviewer 1 and 2. It is written as follows:

*"Figure 1: Location of the Col du Lac Blanc experimental site seen at different scales: (a) general location in France, (b) location within the Grandes Rousses mountain range, (c) details of the study area showing the location of the four AWS surrounding the site and described in Table 1. The blue dashed area on map (c) shows the approximate area covered by the picture in Fig. 2. Contour lines spacing is 100 m for the major lines and 25 m for the minor lines in map b and 50 m and 10 m in map (c). "*

I recommend Table 1 to be merged to Table 2

Table 1 is now merged with Table 2

Line 133-134: The "...from AWS Muzelle and Col are available from winter 2002-2003 and 2010-2011..." is not consistent with Table 2. Check this. Even it is recommended that you remove this as it is already repeated in Table 2.

We thank Reviewer 3 for pointing out this inconsistency. Temperature and snow depth data are available at AWS Muzelle since December 2004 whereas wind data are available since December 2002. We decided to keep the sentence in the manuscript since we believe it is important to explicitly mention in the text that the data from the different stations do not fully cover the period 2000-2016.

Section 2.2.1 Wind speed and direction: I would avoid using indices or names for the sensors (e.g., lines 138, 141, 147, 157) in the text and mention them only in Table 2.

Name of the sensors are now mentioned only in Table 1.

Line 168: replace "get" with "obtain". Correction included